# Mural cells protect the adult brain from hemorrhage but do not control the blood–brain barrier in developing zebrafish

Oguzhan F Baltaci[1,2]*, Andrea Usseglio Gaudi[1,2], Stefanie Dudczig[1,2], Weili Wang[3], Scott Paterson[1,2], Maria Cristina Rondon-Galeano[1,2], Ye-Wheen Lim[3], James Rae[3], Anne Lagendijk[3], Robert G Parton[3,4], Alison Farley[1,2,5], Benjamin M Hogan[1,2,6]*

[1]Organogenesis and Cancer Program, Peter MacCallum Cancer Centre, Melbourne, Australia; [2]Sir Peter MacCallum Department of Oncology, University of Melbourne, Melbourne, Australia; [3]Institute for Molecular Bioscience, The University of Queensland, St Lucia, Australia; [4]Centre for Microscopy and Microanalysis, The University of Queensland, St Lucia, Australia; [5]Walter and Eliza Hall Institute of Medical Research, Melbourne, Victoria, Australia; [6]Department of Anatomy and Physiology, University of Melbourne, Melbourne, Australia

*For correspondence:
oguzhan.baltaci@petermac.org
(OFB);
ben.hogan@petermac.org (BMH)

## eLife Assessment

This **important** study addresses the contribution of pericytes to the organization and permeability control of the zebrafish blood-brain barrier (BBB). By analyzing pdgfrb mutant zebrafish that lack brain pericytes, the authors reveal that the resulting cerebrovascular network is abnormally patterned. Remarkably, however, the barrier retains its restrictive permeability during larval and juvenile stages. More pronounced vascular defects become evident in adults, where localized BBB leakage coincides with hemorrhages and aneurysm formation. Based on **convincing** and beautifully documented imaging data, the authors argue that, unlike what has been reported in rodent systems, pdgfrb-dependent pericytes are not essential for maintaining BBB integrity in the zebrafish brain.

**Abstract** The blood–brain barrier (BBB) protects the brain from circulating metabolites and plays central roles in neurological diseases. Endothelial cells (ECs) of the BBB are enwrapped by mural cells including pericytes and vascular smooth muscle cells (vSMCs) that regulate angiogenesis, vessel stability and barrier function. To explore mural cell control of the BBB, we investigated neurovascular phenotypes in zebrafish *pdgfrb* mutants that lack brain pericytes and vSMCs. As expected, mutants showed an altered cerebrovascular network with mispatterned capillaries. Unexpectedly, mutants displayed no BBB leakage at larval stages of development. This suggests that pericytes and vSMCs are not essential for normal BBB function in developing zebrafish. Instead, we observed juvenile and adult BBB disruption occurring at 'hotspot' focal hemorrhages at large vessel aneurysms. ECs at leakage hotspots showed induction of caveolae on abluminal surfaces and structural defects including basement membrane thickening and disruption. Our work suggests that capillary pericytes primarily regulate cerebrovascular patterning in development and vSMCs of major arteries protect from hemorrhage and BBB breakdown in older zebrafish. The fact that young zebrafish have a functional BBB in the absence of mural cells calls for renewed interrogation of mural cell control of the BBB throughout vertebrate evolution.

## Introduction

Mammalian neurovascular endothelial cells (ECs) have unique functional and structural properties to regulate substance exchange between the blood and the brain, establishing the blood–brain barrier (BBB) (*Zhao et al., 2015*; *Armulik et al., 2011*). Loss of normal BBB function is associated with neuro-pathological states, including neurodegeneration, infection, and cancer (*Zhao et al., 2015*; *Armulik et al., 2011*). Mural cells are vascular support cells that include pericytes and vascular smooth muscle cells (vSMCs) and control vascular development, function and BBB permeability (*Levéen et al., 1994*; *Lindahl et al., 1997*; *Armulik et al., 2010*; *Daneman et al., 2010*; *Eilken et al., 2017*; *Simonavicius et al., 2012*; *Benjamin et al., 1998*). PDGFB/PDGFRβ signaling is required for the development of mural cells (*Levéen et al., 1994*; *Lindahl et al., 1997*; *Hellström et al., 1999*; *Hirschi et al., 1998*; *Soriano, 1994*) and their depletion in PDGFB- or PDGFRβ-deficient rodents causes altered angiogenesis, aneurysm, and vessel leakiness (*Levéen et al., 1994*; *Lindahl et al., 1997*; *Armulik et al., 2010*; *Daneman et al., 2010*; *Ando et al., 2021b*). Strikingly, rodent models of pericyte loss have an open BBB, with free transport of tracer dyes from the blood stream into the brain parenchyma but an otherwise intact vasculature (*Armulik et al., 2010*; *Daneman et al., 2010*). These and other observations have led to a current model whereby pericytes control BBB function by suppressing EC transcytosis, via a mechanism not yet fully understood. Such a mechanism holds significant translational promise to safely 'open' the BBB in therapeutic settings, by discovering ways to disrupt pericyte control of ECs (*Armulik et al., 2010*; *Mäe et al., 2021*).

Studies of the BBB using invertebrates like *Drosophila* and some basal vertebrate fishes (e.g. some *Elasmobranchii* and *Chondrostei*) have suggested that the makeup of the BBB can vary significantly across the phylogeny. These species display a primarily glial cell barrier, rather than an EC barrier (*Paredes-González et al., 2025*; *Limmer et al., 2014*). However, in the modern teleost *Danio rerio* (zebrafish) the BBB cellular composition is more similar to that seen in mammals (*Jeong et al., 2008*; *Xie et al., 2010*; *Quiñonez-Silvero et al., 2020*; *Umans et al., 2017*; *O'Brown et al., 2019*). As in mammals, the zebrafish BBB is controlled by tightly adherent vascular ECs that share many characteristics with the ECs of the mammalian brain. Zebrafish BBB endothelium expresses conserved molecular markers (e.g. *glut1*, *cldn5*, and *mfsd2a*) (*Xie et al., 2010*; *Umans et al., 2017*; *O'Brown et al., 2019*; *Li et al., 2022*; *Vanhollebeke et al., 2015*) and displays similar tight cell–cell junctions as observed at the mammalian BBB (*Jeong et al., 2008*; *Umans et al., 2017*; *Li et al., 2022*). The membrane transport protein Mfsd2a has been suggested to have a conserved role in control of BBB function in zebrafish and mice (*O'Brown et al., 2019*; *Guemez-Gamboa et al., 2015*), likely acting to reduce the transfer of substances between bloodstream and parenchyma by suppressing EC transcytosis (*Ben Zvi et al., 2014*; *Andreone et al., 2017*). Furthermore, during development the mechanisms that control the formation of BBB ECs appear to show a high level of conservation between zebrafish and mice. Recent studies have identified major roles for Wnt–β-catenin signaling, brain EC specific MMPs, Vegfs, and chemokine signaling in early BBB vascular EC development (*Vanhollebeke et al., 2015*; *Eubelen et al., 2018*; *Parab et al., 2023*; *Parab et al., 2021*).

After the initial development of zebrafish BBB vasculature, mural cell populations develop and mature. Transcriptionally, developing pericytes and vSMCs express many genes that are highly conserved with mammals, for example zebrafish pericytes express well known markers such as *pdgfrb*, *kcne4*, and *abcc9* (*Shih et al., 2021*). As in mice, *pdgfrb* mutants fail to form brain pericytes, exhibit loss of vSMCs and display aneurysms as adults (*Ando et al., 2021b*; *Ando et al., 2016*). As well as conserved *pdgfrb* control of mural cell development, Notch signaling (*Ulrich et al., 2016*) also plays a role in mural cell development in zebrafish and in mice, with Notch 1 and 3 controlling the formation of pericytes in mice and Notch 2 and 3 paralogs important in zebrafish (*Wang et al., 2014*; *Ando et al., 2019*). Overall, studies have suggested a high level of conservation of the molecular control of mural cell development, yet how mural cells control BBB function has not been interrogated in detail in zebrafish.

BBB function is first established in zebrafish between 3 days post fertilization (dpf, no barrier) and 5 dpf (established barrier) (*O'Brown et al., 2019*). During this period, developing mural cells (pericytes and vSMCs) are establishing and expanding their interactions with brain ECs (*Ando et al., 2016*; *Ando et al., 2019*; *Ando et al., 2021a*). The first glial cell neurovascular interactions are also observed during this period (*Gall et al., 2025*). Throughout subsequent larval, juvenile and adult development vSMC, pericyte, and glial cell coverage of the vasculature increases (*Ando et al., 2016*; *Gall et al.,*

*2025*; *Chen et al., 2020*). This is similar to mammals, albeit with some differences in the relative timing that lineages arise, and the percentage of coverage of neurovasculature by pericytes and glia may be lower in zebrafish. In mammals, microglia are known to play important roles in the control of the BBB and neurovascular unit (*Mayer and Fischer, 2024*). Zebrafish have a microglial/macrophage population, but it is yet to be explored in detail in the control of the BBB (*Wu et al., 2018*; *Xu et al., 2015*; *Herbomel et al., 2001*) and differences in other innate immune cells in the CNS has been recently reported (*Gaudi et al., 2026*). Overall, while there are some areas remaining to be fully explored, the development, cellular composition and the central role of ECs in the control of the zebrafish BBB is very similar to mammals. As such, zebrafish are accepted as a highly accessible model to study BBB function, as well as mural cell control of the BBB, relevant to mammalian biology (*Quiñonez-Silvero et al., 2020*).

Here, we explored mural cell control of BBB function in zebrafish. We used *pdgfrb* mutants with loss of both pericytes and vSMCs, coupled with carefully optimized intravascular tracer dye injections. We analyzed barrier function from developmental to adult stages and found that at larval stages up to 14 dpf, well after the establishment of the BBB, loss of mural cells does not lead to measurable BBB leakage or loss of BBB integrity. In adult animals, we observed loss of BBB integrity that was consistent with leakage at hotspots, found to be aneurysm associated hemorrhages. These hotspots were also observed in juvenile animals (1 month old) and appeared to increase in number as animals aged. Our observations provide no definitive evidence of BBB leakage at capillary networks that might be consistent with a broad increase in EC transcytosis or loss of barrier integrity independent of focal hemorrhages. This work demonstrates that young animals can have a functional BBB in the absence of mural cells and highlights the central role for the EC in control of barrier function across the zebrafish neurovasculature.

## Results

To investigate the influence of mural cells upon the neurovasculature, we generated a *pdgfrb* mutant (*pdgfrb^uq30bh^*, hereafter *pdgfrb^−/−^*), which possesses an early deletion leading to a frameshift, predicted to cause a premature stop codon and a putative null allele (*Figure 1—figure supplement 1*). Using both the *TgBAC(pdgfrb:EGFP)^uq15bh^* and *TgBAC(abcc9:abcc9-T2A-mCherry)^uom139^* transgenic marker strains, we observed that pericytes were lost in the cerebral central arteries of *pdgfrb* mutants (*Figure 1a*, *Figure 1—figure supplement 1*), as observed in previous studies (*Ando et al., 2021b*; *Ando et al., 2016*). These mutants showed indistinguishable development from their wild-type siblings up to 30 dpf with progressively reduced survival and craniofacial defects thereafter (*Figure 1—figure supplement 1*), consistent with previous studies of *pdgfrb* null loss-of-function mutants (*Ando et al., 2021b*; *Ando et al., 2021a*). To assess how brain vasculature develops without pericytes, we imaged, traced and quantified central arteries in the midbrain (*Figure 1a–c*). At 7 dpf, mutants displayed significantly reduced vascular complexity compared to siblings, with reduced vessel length and branch points (*Figure 1d, e*). This was more severe at 14 dpf, demonstrating the phenotype is progressive (*Figure 1d, e*). Timecourse imaging revealed the reduction in vascular complexity was apparent from 5 dpf (*Figure 1f, g*). These results suggest that the pericytes (as vSMCs mature after 5 dpf; *Ando et al., 2019*) regulate cerebral angiogenesis or remodeling from early in larval development.

We next tested BBB function in mutants at 7 and 14 dpf. We performed intravenous injection of fluorescently conjugated tracers of various molecular weights (1-kDa NHS, 10- and 70-kDa Dextran), and live imaged fish at 2 hours post injection (hpi), the same timing used in *O'Brown et al., 2019*; *Figure 2—figure supplement 1*. 2000-kDa Dextran was co-injected to normalize for the amount of vascular tracer delivered in each animal, and we measured intravascular vs extravascular tracer intensity to assess leakage (see methods for full details). In control animals, quantification confirmed that 2000-kDa tracer remained stably within the vasculature, while 1 and 10 kDa were more readily detected in the brain parenchyma and 70 kDa provided intermediate sensitivity (*Figure 2—figure supplement 1*). This validated the use of 2000-kDa tracer to normalize for amount of tracer injected, as reported elsewhere (*Lim et al., 2026*). To provide additional confidence in our measurements, we also normalized the extravasated tracer intensity to the EC marker *kdrl* (*Tg(kdrl:EGFP)^s843^ or Tg(kdrl:Hsa.HRAS-mCherry)^s916^*). We found no evidence of elevated 10- or 70-kDa Dextran tracer extravasation into the brain parenchyma of *pdgfrb* mutants compared to siblings at either timepoint (7 or 14 dpf) using any of our normalization methods (*Figure 2a–d*, *Figure 2—figure supplement 2*). To

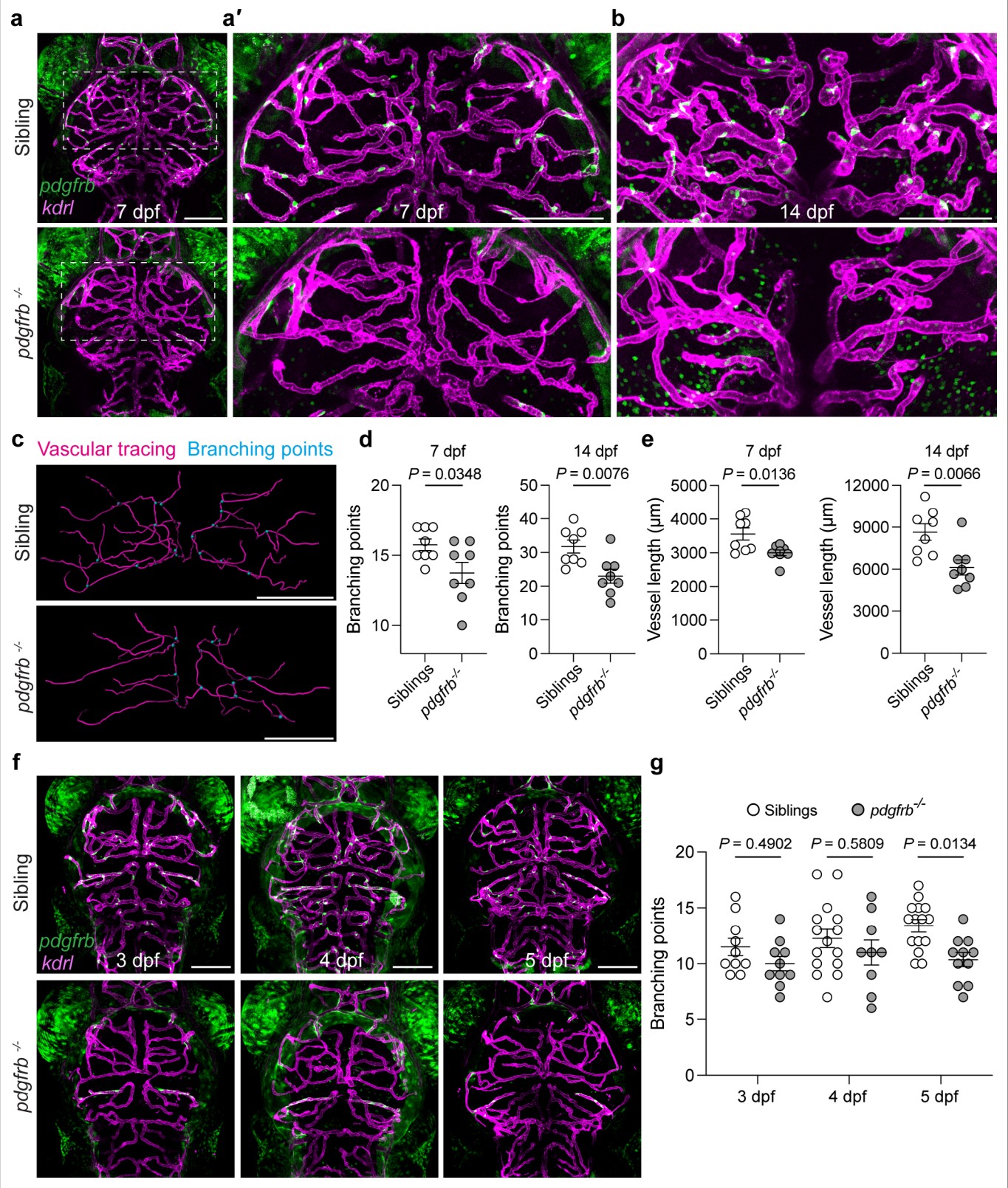

**Figure 1.** Mural cell-deficient larval brain vasculature displays abnormal patterning. Maximum intensity projections of mural cells (*TgBAC(pdgfrb:EGFP)$^{uq15bh}$*) and brain vasculature (*Tg(kdrl:Hsa.HRAS-mCherry)$^{s916}$*) in siblings and *pdgfrb* mutants at 7 (**a**) and 14 dpf (**b**). (**a'**) shows enlarged view of the white box in (**a**) highlighting the midbrain central arteries. (**c**) Representative images of vascular tracing at 7 dpf using Imaris 10.1. Lumenized blood vessels branching from the middle mesencephalic central arteries in the midbrain were traced. (**d–e**) Quantification of midbrain

*Figure 1 continued on next page*

*Figure 1 continued*

branching points (**d**) and vessel length (**e**). Data are mean ± SEM, *n* = 8 per group, unpaired *t*-tests. (**f**) Maximum intensity projections of mural cells (*TgBAC(pdgfrb:EGFP)^uq15bh*) and brain vasculature (*Tg(kdrl:Hsa.HRAS-mCherry)^s916*) in siblings and *pdgfrb* mutants at 3–5 dpf. (**g**) Quantification of branching points of midbrain central arteries demonstrating the first detectable vessel patterning phenotype at 5 dpf. Data are mean ± SEM, *n* = 10 per group at 3 dpf, *n* = 15 sibling and 9 *pdgfrb^−/−* at 4 dpf, *n* = 15 sibling and 11 *pdgfrb^−/−* at 5 dpf, two-way analysis of variance (ANOVA) followed by multiple comparisons with Tukey's correction. (**a, a', b, c, f**) Dorsal views, anterior up, scale bars: 100 µm.

The online version of this article includes the following figure supplement(s) for figure 1:

**Figure supplement 1.** *uq30bh* is a predicted null allele of *pdgfrb* causing brain pericyte deficiency.

confirm that our quantification methods were sufficiently sensitive to detect BBB leakage, we assessed 10- and 70-kDa tracer leakage before and after the establishment of BBB function (at 3 and 7 dpf; *O'Brown et al., 2019*). With both 10- and 70-kDa tracers, we could detect an open BBB at 3 dpf that was closed at 7 dpf (*Figure 2e, f*). Together, these results show that we can accurately measure BBB leakage and integrity in zebrafish and that based on these measures neither pericytes nor vSMCs are necessary for BBB integrity during larval stages.

To investigate how mural cells control cerebral vasculature at later stages of life, we tissue-cleared brains of adult stage *pdgfrb* mutants and siblings (*Matsumoto et al., 2019*) and performed whole brain imaging. We confirmed loss of pericytes and vSMCs at this stage using *TgBAC(pdgfrb:EGFP)^uq15bh* and *TgBAC(acta2:EGFP)^uq17bh* (*Figure 3a, b*). We examined a consistent region of the left optic tectum (midbrain) and detected decreased branching of capillaries and increased diameter of major vessels (*Figure 3d–f*). The observed vessel dilation was restricted to larger caliber arteries that would normally be invested with vSMCs (*Figure 3—figure supplement 1*) and was not seen in capillaries in this region of the brain (*Figure 3e, f*). The loss of vSMCs was observable from the earliest stages of vSMC development and artery dilation was observed by 21 dpf, consistent with previous studies (*Ando et al., 2021b*; *Figure 3c*, *Figure 3—figure supplement 1*). This suggests that major vessel dilation seen in *pdgfrb* mutants is due to loss of vSMCs and that reduced branching of capillaries is due to loss of pericytes (given the timing observed in *Figure 1*).

To investigate BBB integrity at later stages, we injected 3-month-old adult zebrafish with 10-kDa Dextran. Brightfield imaging of vibratome-sectioned brains revealed severe aneurysms across large caliber vessels (*Figure 4a*) with blood pooling that was potentially consistent with vessel ruptures (*Figure 4c*) as previously reported (*Ando et al., 2021a*). Furthermore, we detected parenchymal accumulation of tracer dye closely associated with large caliber vessel aneurysms (*Figure 4b, c*). To quantify the distribution of parenchymal tracer dye from deep (around major artery aneurysms) to superficial (capillary beds) brain regions, we measured fluorescence intensity in 10 brain regions approximately 100 µm apart from medial to lateral locations. This demonstrated accumulation of tracer dye in medial regions associated with aneurysm in mutants (*Figure 4d, e*). These animals showed little evidence of leakage in capillary beds, with marginally higher tracer intensity in mutants than controls in lateral locations that could be due to diffusion from medial locations with high concentrations of tracer (*Figure 4e*). To better understand this extravasated tracer accumulation, we conducted whole brain imaging after tracer injection in 5-month-old adults. Interestingly, we observed highly localized 70-kDa tracer accumulation in regions that could be considered leakage 'hotspots' at large caliber vessel aneurysms (*Figure 5a*; *Video 1* and *Video 2*).

As 10-kDa tracer showed higher sensitivity than 70-kDa tracer in our control experiments (*Figure 2—figure supplement 1*), we examined 10-kDa tracer injections at earlier stages and found that hotspots were readily detectable from as early as 1 month using this lower molecular weight tracer (*Figure 5b*). The hotspots were not readily detectable using 70-kDa Dextran in 2-month-old zebrafish, consistent with increased sensitivity of 10-kDa tracers (*Figure 5—figure supplement 1*). Interestingly, the quantification of hotspots in whole cleared brains at 1 and 5 months of age showed that the number increased with aging, suggesting a progressive loss of BBB integrity at focal sites of leakage (*Figure 5c*). To further understand the nature of leakage hotspots, we examined red blood cells in *pdgfrb* loss-of-function adults. We used *pdgfrb* Crispant knockouts, which we validated by finding that F0 CRISPR consistently reproduced the adult mutant phenotype (*Figure 5—figure supplement 1*). Using the *Tg(gata1:DsRed)^sd2* transgenic line to label erythrocytes, together with tracer injection, we found that knockout animals showed tracer accumulation concomitant with extravasated red blood cells at hotspots (*Figure 5d*). This identified hotspots as sites of focal hemorrhage. Notably, these sites

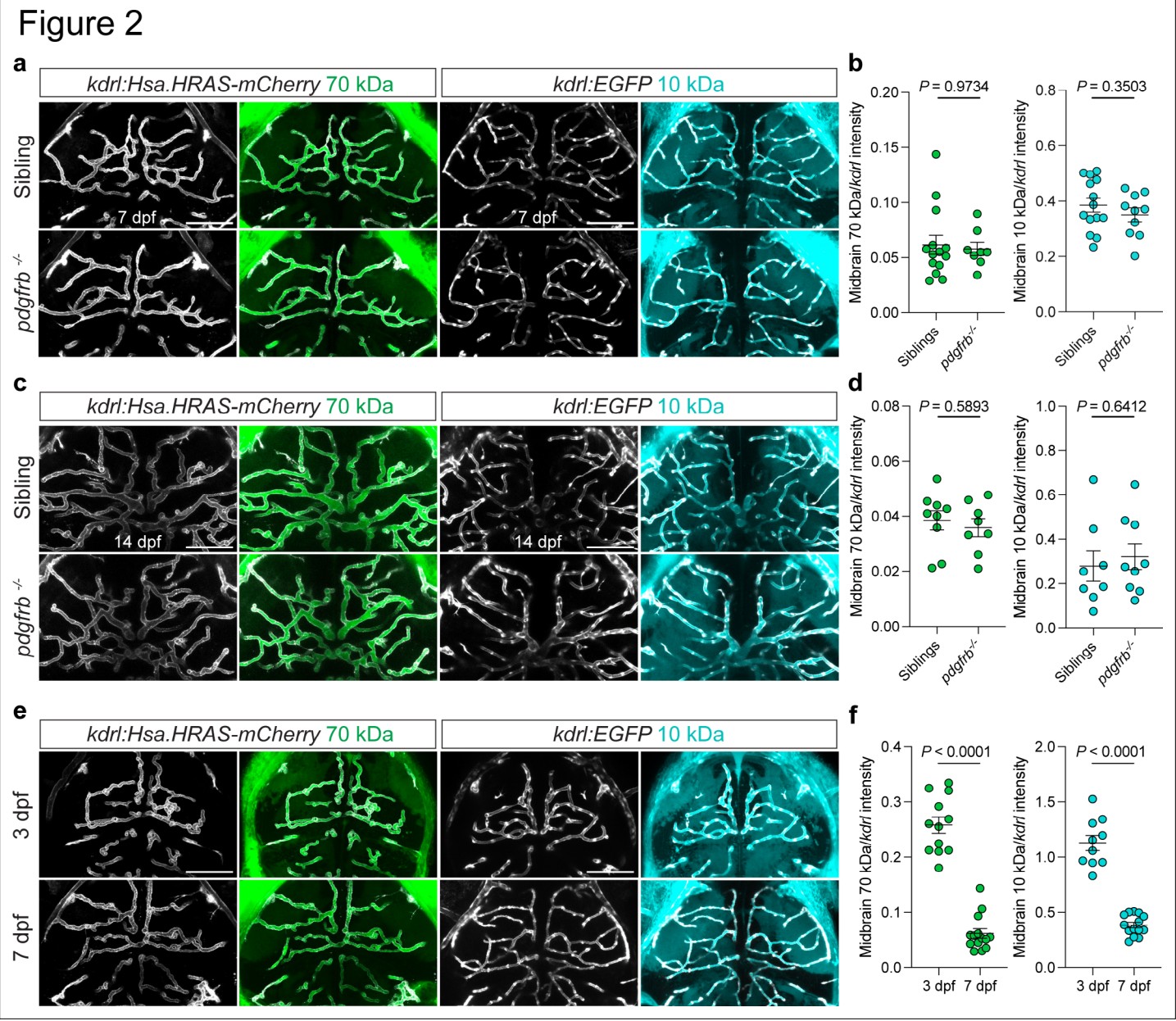

**Figure 2.** BBB integrity is maintained in *pdgfrb* mutants during larval stages. (**a, c, e**) Fluorescent tracer leakage assays in the midbrain of zebrafish larvae using 70-kDa Dextran–Fluorescein or 10-kDa Dextran–Alexa Fluor 647, visualized with vascular reporters (gray) *Tg(kdrl:Hsa.HRAS-mCherry)^s916* and *Tg(kdrl:EGFP)^s843*. Siblings and *pdgfrb* mutants at 7 dpf (**a**) and 14 dpf (**c**) show no differences in tracer extravasation. Wild-type larvae at 3 and 7 dpf (**e**) serve as a positive control, illustrating reduced tracer leakage by 7 dpf. Dorsal views, anterior up, scale bars: 100 µm. (**b, d, f**) Quantification of midbrain parenchymal 70- and 10-kDa dextran fluorescence intensity normalized to vascular *kdrl* reporter intensity. Data are mean ± SEM. At 7 dpf (**b**): *n* = 14 siblings and 8 *pdgfrb^−/−* for 70 kDa, and *n* = 14 siblings and 10 *pdgfrb^−/−* for 10 kDa. At 14 dpf (**d**): *n* = 9 siblings and 8 *pdgfrb^−/−* for 70 kDa, and *n* = 8 siblings and 9 *pdgfrb^−/−* for 10 kDa. For the positive control at 3 and 7 dpf (**f**): *n* = 12 (3 dpf) and 14 (7 dpf) for 70 kDa, and *n* = 10 (3 dpf) and 14 (7 dpf) for 10 kDa. Unpaired *t*-tests for 10-kDa dextran comparisons at 3 vs 7 dpf and for sibling vs *pdgfrb^−/−* comparisons at 7 and 14 dpf, and for 70-kDa dextran at 14 dpf. Mann–Whitney *U* tests for 70-kDa dextran comparisons at 3 vs 7 dpf and for sibling vs *pdgfrb^−/−* comparisons at 7 dpf due to non-normal data distribution.

The online version of this article includes the following figure supplement(s) for figure 2:

**Figure supplement 1.** Controls for size-dependent tracer extravasation and distribution at larval and adult stages.

**Figure supplement 2.** Tracer assays using an independent normalization method confirm that BBB integrity is maintained in *pdgfrb* mutant larvae.

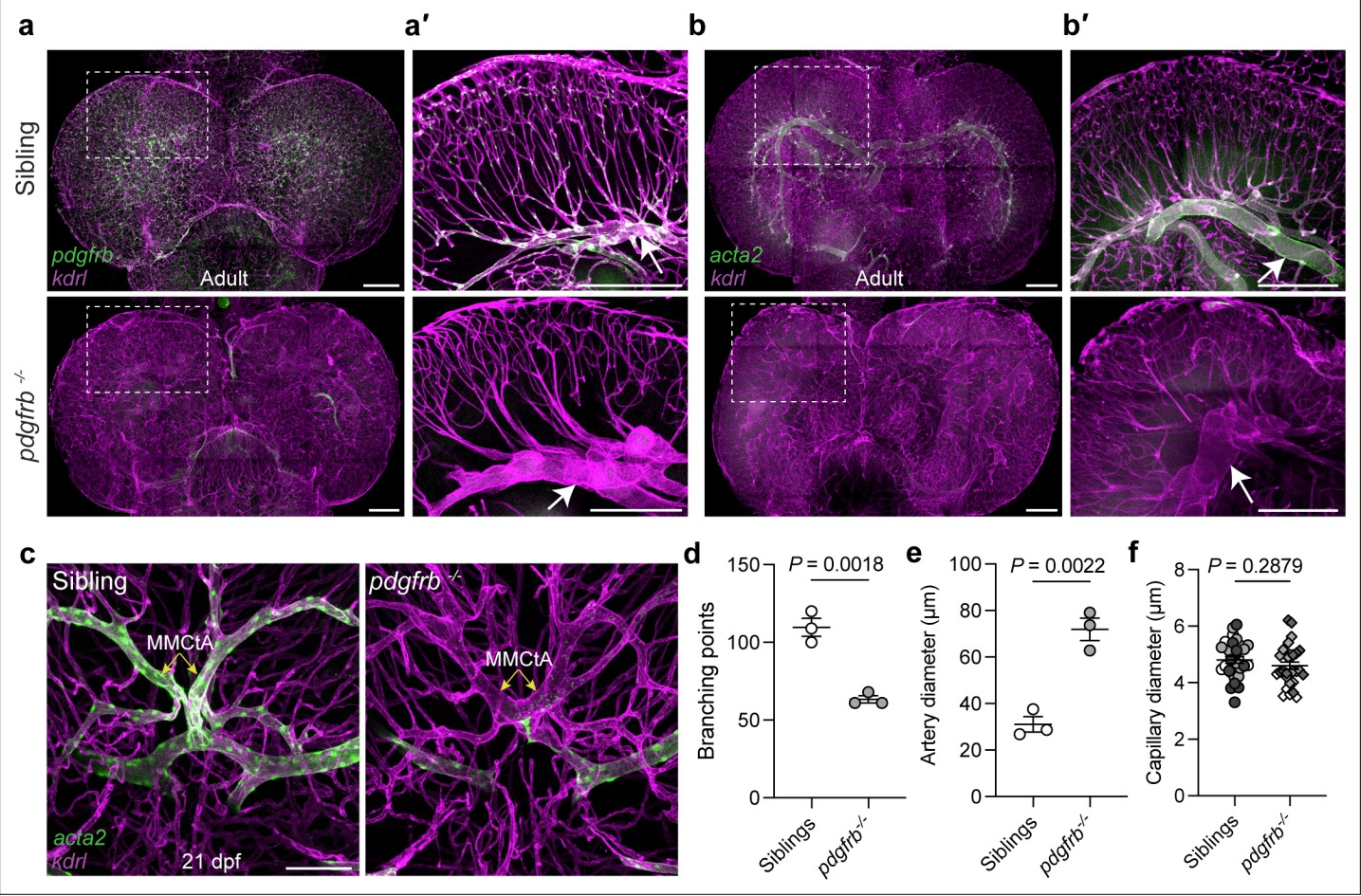

**Figure 3.** Adult *pdgfrb* mutants lack mural cells and display dilated arteries. Confocal projections of whole brain imaging in siblings and *pdgfrb* mutants at 5 months of age, showing brain vasculature using *Tg(kdrl:Hsa.HRAS-mCherry)^s916^* and mural cells using *TgBAC(pdgfrb:EGFP)^uq15bh^* (**a**) as a global marker and *TgBAC(acta2:EGFP)^uq17bh^* (**b**) as a vSMC marker. Dorsal views, anterior up, scale bars: 250 µm. (**a', b'**) Subset of z-slices from the whole brain imaging in (**a**) and (**b**) (white boxes) indicating mural cell loss and abnormal capillary network patterning. 100-µm-thick maximum intensity projections (MIPs) were generated using the continuation of the left middle mesencephalic central artery (MMCtA, arrow) as an anatomical landmark. Scale bars: 250 µm. (**c**) Confocal projections from whole brain imaging in sibling and *pdgfrb* mutants at 21 dpf. Siblings show *acta2:EGFP*-positive vSMC coverage along MMCtAs (yellow arrows). *pdgfrb* mutants have dilated MMCtAs with absent vSMCs. Scale bar: 100 µm. Quantification of capillary branching points (**d**), artery diameter (**e**), and capillary diameter (**f**) using the area shown in (**a'**). Data are mean ± SEM, *n* = 3 animals per group. Artery diameter was quantified by averaging 5 points per animal, and capillary diameter was quantified by measuring 10 capillary diameter per animal. Each animal marked with a different color in **f**. Unpaired *t*-tests for branching points and artery diameter, nested *t*-test for capillary diameter.

The online version of this article includes the following figure supplement(s) for figure 3:

**Figure supplement 1.** Development of *acta2:EGFP*-positive vSMCs in brain vasculature of siblings and *pdgfrb* mutants.

were not detected at capillaries (*Figure 5e*). Taken together, this shows that as mutants age, loss of BBB integrity is associated with an increasing number of hotspot sites of focal hemorrhage, that are found along aneurysmal vessels.

The quantification of tracer dye intensity at 2 hpi in adult mutants from medial (major vessels) to lateral (capillaries) locations indicated leakage most closely associated with hotspot focal hemorrhages. However, the intensity of tracer dye found in the lateral capillary regions was measurably higher than in control animals (*Figure 4e*). To test if this was likely due to progressive diffusion of tracers from medial to lateral locations or if there may be some local leakage at capillaries, we examined 10-kDa tracer leakage at 0.5 and 6 hpi. While hotspots could be detected at 0.5 hpi, we detected little tracer accumulation in medial locations and no evidence of parenchymal tracer leakage at lateral capillary locations (*Figure 5f, h*, *Figure 5—figure supplement 2*). However, after 6 hr of diffusion of tracer within the brain, we clearly observed tracer at both medial and lateral locations, with higher

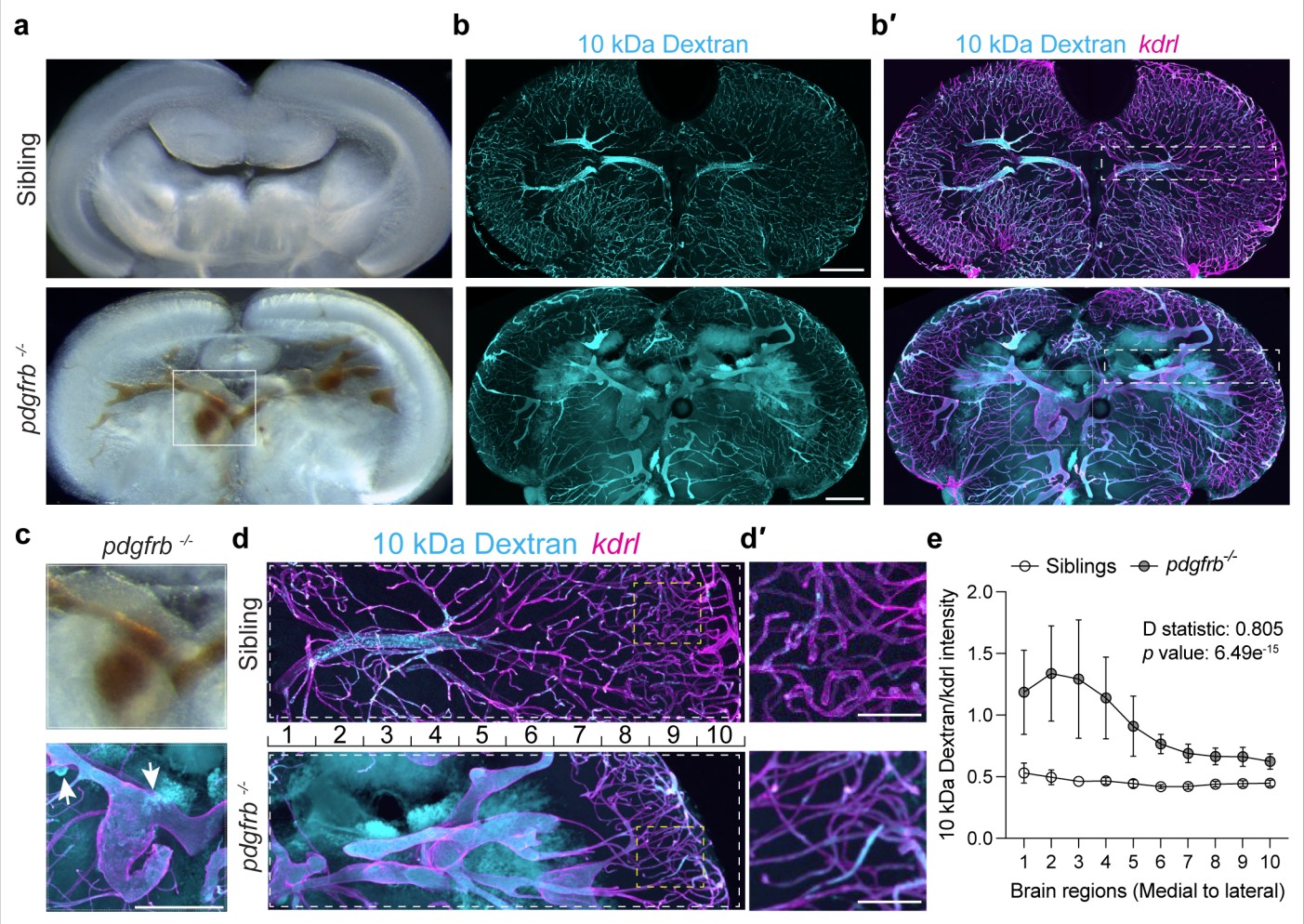

**Figure 4.** Adult *pdgfrb* mutants display aneurysms and vascular integrity defects. (**a**) Representative images of coronal midbrain sections obtained from 3-month-old sibling and *pdgfrb* mutants. (**b**) Fluorescent tracer leakage assays in 3-month-old siblings and *pdgfrb* mutants (2 hpi). 10-kDa Dextran–Cascade Blue (**b**) was used to assess extravasation, and *Tg(kdrl:Hsa.HRAS-mCherry)*$^{s916}$ was used to label the brain vasculature (merge in **b'**). Scale bar: 250 µm. (**c–c'**) Enlarged regions from (**a**) and (**b'**) (dotted box) displaying blood and tracer accumulation (arrows) in *pdgfrb* mutants. Scale bar: 250 µm. (**d**) Enlarged regions from (**b**) (dashed box) displaying vasculature and tracer leakage from medial (arterial region) to lateral (capillary region) regions. Area was divided into 10 regions (~100 µm intervals) for quantification. (**d'**) Enlarged view of yellow box in (**d**) showing capillary beds with intravascular tracer. Scale bar: 50 µm. (**e**) Quantification of parenchymal 10-kDa tracer intensity (medial to lateral) normalized to vascular *kdrl* intensity. Data are mean ± SEM, each point represents average normalized tracer intensity per brain region, *n* = 5 sibling and 4 *pdgfrb*$^{-/-}$, two-sample Kolmogorov–Smirnov test.

intensity in medial locations (*Figure 5g, i*; *Figure 5—figure supplement 2*). While we cannot rule out the possibility of slow leakage of tracer at capillaries occurring concurrently with leakage at hemorrhages, these observations seem consistent with a primary source of leakage at focal hemorrhages, followed by diffusion of tracer throughout the parenchyma.

Pericytes are proposed to regulate the BBB by suppressing transcytosis through brain capillary ECs (*Armulik et al., 2010*; *Daneman et al., 2010*). This model is built on the observation by electron microscopy of the induction of increased numbers of vesicular structures and caveolae in leaky pericyte-deficient vessels (*Armulik et al., 2010*; *Daneman et al., 2010*). To investigate if the loss of BBB integrity is associated with ultrastructural changes, we performed electron microscopy. We imaged large caliber vessels and capillaries (*Figure 6a, b*, *Figure 6—figure supplement 1*) and examined cellular junctions, basement membrane and caveolae (defined as uncoated vesicular profiles of <100 nm diameter). While tight junctions between ECs remained intact, caveolae numbers along large vessel aneurysms in *pdgfrb* mutants were significantly increased (*Figure 6a''–c*). This increase was almost exclusive to the abluminal side of the ECs (*Figure 6c*). This abluminal accumulation of caveolae may be more associated with changes in the mechanical state of ECs (as caveolae are highly

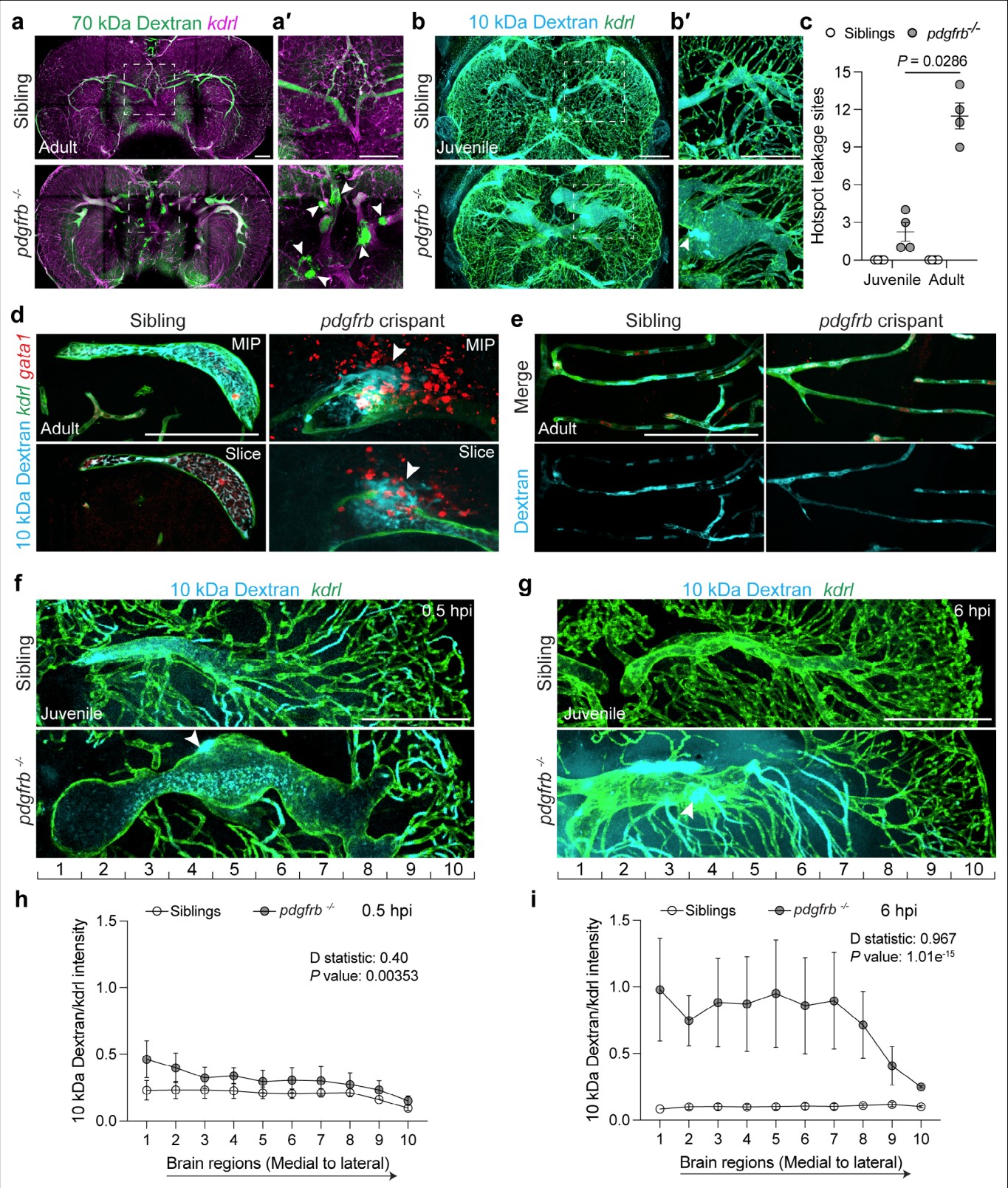

**Figure 5.** Adult and juvenile *pdgfrb* mutants display vessel rupture and tracer accumulation at aneurysm hotspots. (**a–a′**) Fluorescent tracer leakage assays in 5-month-old sibling and *pdgfrb* mutants. 70-kDa Dextran–Fluorescein was detected outside the vessels marked by *Tg(kdrl:Hsa.HRAS-mCherry)*[s916] at aneurysm hotspots (*pdgfrb* mutant, arrowheads) shown in white box (magnified in **a′**). (**b–b′**) Fluorescent tracer leakage assays in 1-month-old sibling and *pdgfrb* mutants. 10-kDa Dextran–Alexa Fluor 647 was detected outside the vessels marked by *Tg(kdrl:EGFP)*[s843] at aneurysm

*Figure 5 continued on next page*

*Figure 5 continued*

hotspots (arrowhead) shown in white box (magnified in **b'**). (**a–b**) Brains were collected at 2 hpi, dorsal views, anterior is up, scale bars: 200 µm. (**c**) Quantification of hotspot leakage sites per animal in the midbrain region showing increased numbers in older animals. Data are mean ± SEM, n = 4 per group, unpaired *t*-test. High-resolution imaging of artery (**d**) and capillary zones (**e**) in brain regions of 10-week-old adult *pdgfrb* crispants and uninjected siblings after tracer injection (2 hpi). 10-kDa Dextran–Alexa Fluor 647 injected, blood vessels (*Tg(kdrl:EGFP)*[s843]) and red blood cells (*Tg(gata1:DsRed)*[sd2]) are shown in maximum intensity projections (MIPs) and single Z-sections. Arrowheads indicate examples of hotspots. Capillary zones shown in MIPs (all channels) and single 10-kDa Dextran channel lack hotspots. Scale bar: 100 µm. Fluorescent tracer leakage assays in 1-month-old sibling and *pdgfrb* mutants shown in confocal projections of coronal midbrain sections collected at 0.5 hpi (**f**) or 6 hpi (**g**). 10-kDa Dextran–Alexa Fluor 647 was used to assess extravasation, and *Tg(kdrl:EGFP)*[s843] was used to label the brain vasculature. The area was divided into 10 regions (~70 µm intervals) for quantification (see *Figure 5—figure supplement 2* for full section image). Arrowheads indicate examples of hotspots. Scale bar: 200 µm. (**h, i**) Quantification of parenchymal 10-kDa dextran intensity (medial to lateral) normalized to vascular *kdrl* intensity. Data are mean ± SEM, each point represents average normalized tracer intensity per brain region, n=3 sibling and 5 *pdgfrb*[−/−] for 0.5 hpi, and *n* = 3 per group for 6 hpi, two-sample Kolmogorov–Smirnov test.

The online version of this article includes the following figure supplement(s) for figure 5:

**Figure supplement 1.** Arterial aneurysms in juvenile *pdgfrb* mutants and additional control data.

**Figure supplement 2.** Coronal midbrain sections corresponding to *Figure 5f, g*.

regulated by membrane tension *Sinha et al., 2011*; *Del Pozo et al., 2021*) than changes in intracellular transport. Furthermore, we detected thickening of basement membrane (BM) (*Figure 6d*), gaps in the BM and fluid accumulation outside the vessel wall at aneurysms in mutants. We found no obvious difference in the caveolae density in capillary ECs of *pdgfrb* mutants compared with siblings (*Figure 6—figure supplement 1*). Thus, leakage hotspots at aneurysms are associated with a loss of the normal flattened morphology of ECs, caveola induction and disruption of the BM, all factors that could contribute to or be observed concurrent with vessel wall rupture.

## Discussion

Brain vasculature is endowed with mural cells that control vascular development and stability (*Levéen et al., 1994*; *Lindahl et al., 1997*; *Ando et al., 2021b*; *Rajan et al., 2020*). Studies in rodents have shown that pericytes regulate BBB integrity, with knockout of *Pdgfb* or *Pdgfrb* leading to vessel dilations and brain hemorrhages from early in development (*Levéen et al., 1994*; *Lindahl et al., 1997*). However, hypomorphic mutant alleles impacting *Pdgfb* or *Pdgfrb* show increased leakage of the BBB, with the severity of the leakage phenotype correlating with the expressivity of the pericyte phenotype (*Armulik et al., 2010*; *Daneman et al., 2010*). BBB leakage is associated with increased vesicular structures in ECs and the 'open' BBB state in the absence of mural cells is thought to be caused by hotspots of increased EC transcytosis

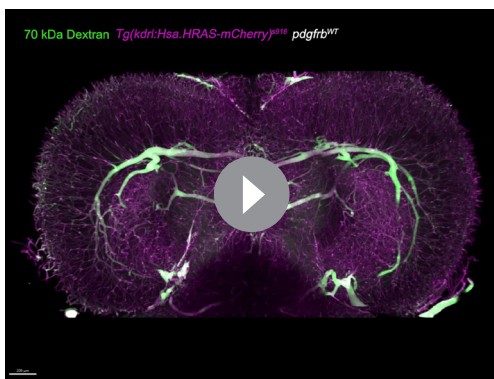

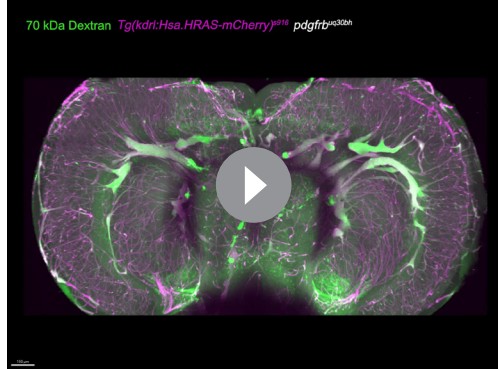

**Video 1.** 3D reconstruction of a wild-type CUBIC-cleared zebrafish midbrain at 5 months of age. Intravenously injected 70-kDa Dextran–Fluorescein (green) remained intact in the brain vasculature labeled by *Tg(kdrl:Hsa.HRAS-mCherry)*[s916].

https://elifesciences.org/articles/104061/figures#video1

**Video 2.** 3D reconstruction of a *pdgfrb* mutant CUBIC-cleared zebrafish midbrain at 5 months of age. Intravenously injected 70-kDa Dextran–Fluorescein (green) was detected outside the vessels marked by *Tg(kdrl:Hsa.HRAS-mCherry)*[s916] at aneurysm hotspots (white boxes).

https://elifesciences.org/articles/104061/figures#video2

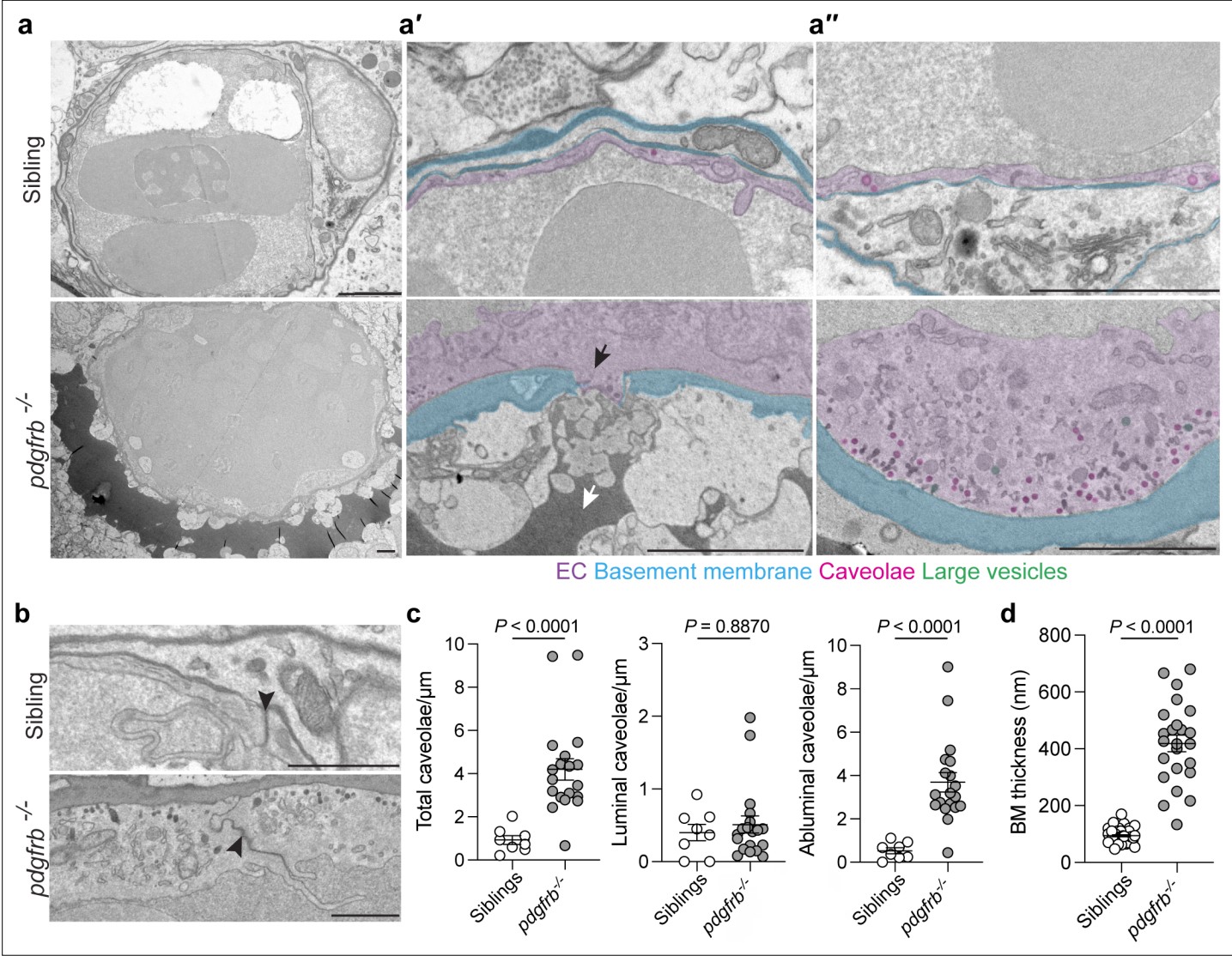

**Figure 6.** Adult *pdgfrb* mutants display structural endothelial cell defects and vessel rupture. (**a–a″**) Transmission electron microscopy images of sectioned adult zebrafish brain vessels at 5 months of age. *pdgfrb* mutants show basement membrane thickening and breakdown (black arrow), serum accumulation outside the vessels (white arrow) (**a′**), increased abluminal endothelial caveolae (**a″**). Pseudocolors shown are ECs (purple), basement membrane (cyan), caveolae (magenta) and vesicles larger than 100 nm (green). Scale bars: 2 µm. (**b**) Magnified regions from (**a**) showing intact tight junctions (arrowheads) in both siblings and *pdgfrb* mutants. Scale bars: 1 µm. (**c**) Quantification of endothelial caveolae. Caveolae were defined as uncoated spherical profiles <100 nm in diameter and scored as luminal or abluminal (see methods for details). Measurements were made for *n* = 3 vessels for each group and a total of 8 different cellular regions measured for siblings and 19 for mutants. Data are mean ± SEM and are not distributed normally, Mann–Whitney *U* test was applied. (**d**) Quantification of basement membrane thickness, which was scored in 6 different regions per vessel with *n* = 8 vessels in siblings 4 in *pdgfrb* mutants across *n* = 3 animal per group. Data are mean ± SEM, unpaired *t*-test. Data are mean ± SEM and are not distributed normally, Mann–Whitney *U* test was applied.

The online version of this article includes the following figure supplement(s) for figure 6:

**Figure supplement 1.** Adult *pdgfrb* mutants maintain normal endothelial ultrastructure in capillaries.

(***Armulik et al., 2011***; ***Tjakra et al., 2019***). Other potentially relevant pathways in ECs that could be altered by pericytes include altered cell–cell adhesion (e.g. Cldn5; ***Mäe et al., 2021***), control of Mfsd2a (***O'Brown et al., 2019***; ***Andreone et al., 2017***), Netrin1-Unc5B signaling (***Boyé et al., 2022***), or vitronectin regulation of integrin receptors (***Ayloo et al., 2022***), but direct links are yet to be established. Some studies using other models to deplete pericytes in mice have reported loss of pericytes without vessel leakage. For example, optical or chemical ablation of pericytes in the brain or retina of mice can be achieved without compromising the integrity of BBB or blood–retinal barrier (***Berthiaume***

*et al., 2022*; *Park et al., 2017*), however these methods may induce inflammatory or compensatory mechanisms that also influence local vasculature. Taken together, these studies show that mural cells and in particular pericytes control the BBB in mammals, yet the precise downstream mechanisms by which pericytes regulate EC function require further study.

We show using zebrafish that mural cells are involved in cerebrovascular patterning in larval and juvenile animals, but these vessels maintain BBB function in the absence of mural cells. While the zebrafish BBB has been shown to be established between 3 and 5 dpf, our leakage assays at 7 and 14 dpf in mural cell-deficient *pdgfrb* mutants indicate intact BBB function and vessel integrity. This is unlikely to be due to technical issues with detecting leakage, as multiple measures and methods of tracer normalization produced consistent results. Furthermore, we can readily detect the 'open' BBB before it closes during development (*Figure 2e, f*). This demonstrates that a functional vertebrate BBB can exist in the absence of pericytes. Importantly, while other species have a glial cell BBB (e.g. *Drosophila*, some sharks and sturgeon), the zebrafish is well established to have an EC barrier and the glial cell coverage at the stages analyzed here is far from complete (*Gall et al., 2025*), making glial cell compensation for a loss of pericytes unlikely. There are other evolutionary differences (e.g. developmental timing, innate immune differences *Gaudi et al., 2026*; *Renshaw and Trede, 2012*) between zebrafish and mammals and our observation of intact BBB function without mural cells could represent an additional divergence. However, the high level of conservation of developmental pathways in BBB ECs and the very similar transcriptional signatures of both neurovascular ECs and mural cells between species (*Quiñonez-Silvero et al., 2020*; *Shih et al., 2021*; *Ando et al., 2019*; *Vanlandewijck et al., 2018*) may make this surprising. Once additional mechanisms of pericyte–EC crosstalk are established in mice and zebrafish, deeper mechanistic comparative studies will help to better understand if our observations identify major differences in BBB control between these organisms or not.

Ultimately, in older mutants from 1 month of age onwards, BBB integrity defects are observable with major arteries progressively harboring aneurysms and focal hemorrhages (leakage hotspots). These aneurysms are likely due to the loss of vSMCs seen in our *pdgfrb* mutants and other zebrafish *pdgfrb* mutant models (*Ando et al., 2021b*; *Ando et al., 2021a*). Adult mutant hotspots show extravascular erythrocytes and tracer dye, demonstrating that focal hemorrhages are a cause of vessel leakage. Similar to this, in *Pdgfb* or *Pdgfrb* mutant mice with strong phenotypic expressivity, hemorrhages are also reported (although not in mild or hypomorphic mutants used in BBB studies) (*Levéen et al., 1994*; *Lindahl et al., 1997*; *Soriano, 1994*). These observations suggest a highly conserved role for mural cells in the regulation of aneurysm and hemorrhage in mice and zebrafish. In our zebrafish model, BBB leakage in juvenile and adult animals appears to only be observed once hemorrhages or hotspots are present. Coupled with the fact that larval zebrafish can have no mural cells, but intact barrier function, we suggest that any roles for mural cells in BBB integrity beyond control of aneurysm and hemorrhage are likely to be minor roles in this setting. It remains possible that there is slow undetectable capillary leakage, but future models that specifically lack capillary pericytes and not vSMCs will be needed to test this. Understanding the nature of hotspot leakage more broadly in various mammalian and non-mammalian models of mural cell deficiency would seem likely to uncover further insights into BBB regulation. Deeper understanding of the fundamental biology of the BBB is still needed and will pave the way for increasingly sophisticated efforts to target the BBB in disease in the future.

## Materials and methods

### Key resources table

| Reagent type (species) or resource | Designation | Source or reference | Identifiers | Additional information |
|---|---|---|---|---|
| Gene (*Danio rerio*) | *pdgfrb* | Ensembl | ZDB-GENE-030805-2 | |
| Gene (*Danio rerio*) | *abcc9* | Ensembl | ZDB-GENE-050517-23 | |
| Gene (*Danio rerio*) | *acta2* | Ensembl | ZDB-GENE-030131-1229 | |
| Gene (*Danio rerio*) | *slc45a2* | Ensembl | ZDB-GENE-050208-97 | Albino |

Continued

| Reagent type (species) or resource | Designation | Source or reference | Identifiers | Additional information |
|---|---|---|---|---|
| Genetic reagent (*Danio rerio*) | *pdgfrb*<sup>uq30bh</sup> | This paper | ZDB-LAB-100302-2 | Materials and methods |
| Genetic reagent (*Danio rerio*) | *TgBAC(pdgfrb:EGFP)*<sup>uq15bh</sup> | PMID:28459441 | ZDB-ALT-180306-11 | |
| Genetic reagent (*Danio rerio*) | *Tg(kdrl:Hsa.HRAS-mCherry)*<sup>s916</sup> | PMID:30204931 | ZDB-ALT-090506-2 | |
| Genetic reagent (*Danio rerio*) | *Tg(kdrl:EGFP)*<sup>s843</sup> | PMID:16251212 | ZDB-ALT-050916-14 | |
| Genetic reagent (*Danio rerio*) | *Tg(gata1:DsRed)*<sup>sd2</sup> | PMID:14608381 | ZDB-ALT-051223-6 | |
| Genetic reagent (*Danio rerio*) | *TgBAC(acta2:EGFP)*<sup>uq17bh</sup> | PMID:28459441 | ZDB-ALT-180306-12 | |
| Genetic reagent (*Danio rerio*) | *Tg(5xUAS:RFP)*<sup>nkuasrfp1a</sup> | PMID:18202183 | ZDB-ALT-080528-2 | |
| Genetic reagent (*Danio rerio*) | *TgBAC(abcc9:abcc9-T2A-mCherry)*<sup>uom139</sup> | This paper | ZDB-LAB-100302-2 | Materials and methods |
| Genetic reagent (*Danio rerio*) | *TgBAC(pdgfrb:gal4FF)*<sup>uom140</sup> | This paper | ZDB-LAB-100302-2 | Materials and methods |
| Recombinant DNA reagent | pRedET (plasmid) | Gene Bridges | | |
| Recombinant DNA reagent | pCS2-T2A-mCherry-KanR (plasmid) | This paper | ZDB-LAB-100302-2 | Materials and methods |
| Recombinant DNA reagent | *abcc9* BAC clone | BACPAC | CH211-58C15 | |
| Recombinant DNA reagent | *pdgfrb:gal4FF* BAC clone | PMID:26952986 | | |
| Sequence-based reagent | T2A_mcherry_F | This paper | PCR primers | atggagcaggaggacggcctgtttgcatcttttgtcaaagccgacatgGAGGGCAGAGGAAGTCTGCTA |
| Sequence-based reagent | T2A_mcherry_F | This paper | PCR primers | aaaatggcttttattgatctgttaaggccaaaagtggtgtaaagtggggaTCAGAAGAACTCGTCAAGAAGGCG |
| Sequence-based reagent | gRNA_uq30bh | This paper | gRNA | GATGGTGACTAAGACGCGA |
| Sequence-based reagent | Forward genotyping primer for *uq30bh* allele | This paper | PCR primers | CTTCCTTAGATCCTGACGTGTG |
| Sequence-based reagent | Reverse genotyping primer for *uq30bh* allele | This paper | PCR primers | TATTGATGGGTTCGTCACCAG |
| Sequence-based reagent | Dr.Cas9.PDGFRB.1.AA | IDT | crRNA | GATGGTGACTAAGACGCGAG |
| Sequence-based reagent | Dr.Cas9.PDGFRB.1.AB | IDT | crRNA | CTCGGTGCACACATAAACCC |
| Sequence-based reagent | PDGFRB.1.AB_F | This paper | PCR primers | GACGAGAACATCCCAGACTTTC |
| Sequence-based reagent | PDGFRB.1.AB_R | This paper | PCR primers | GCGTGTAAACAAATCCTAACGG |
| Chemical compound, drug | Phenylthiourea | Sigma-Aldrich | P7629 | |
| Chemical compound, drug | Low-melt agarose | Bio-Rad | 1613112 | |

*Continued on next page*

| Reagent type (species) or resource | Designation | Source or reference | Identifiers | Additional information |
|---|---|---|---|---|
| | | *Continued* | | |
| Chemical compound, drug | Paraformaldehyde | Sigma-Aldrich | P6148 | |
| Chemical compound, drug | Glutaraldehyde | ProSciTech | C002 | |
| Chemical compound, drug | RapiClear 1.52 | SUNJin Lab | RC152001 | |
| Chemical compound, drug | Alexa Fluor 405 NHS Ester (Succinimidyl Ester) | Thermo Fisher | A30000 | |
| Chemical compound, drug | Dextran, Cascade Blue, 10,000 MW, Anionic, Lysine Fixable | Thermo Fisher | D1976 | |
| Chemical compound, drug | Dextran, Alexa Fluor 647; 10,000 MW, Anionic, Fixable | Thermo Fisher | D22914 | |
| Chemical compound, drug | Dextran, Fluorescein, 70,000 MW, Anionic, Lysine Fixable | Thermo Fisher | D1822 | |
| Chemical compound, drug | Dextran, Tetramethylrhodamine, 2,000,000 MW, Lysine Fixable | Thermo Fisher | D7139 | |
| Software, algorithm | Fiji (ImageJ) | PMID:22743772 | RRID:SCR_002285 | |
| Software, algorithm | Imaris 10.1 | Oxford Instruments | RRID:SCR_007370 | |
| Software, algorithm | GraphPad Prism | GraphPad | RRID:SCR_002798 | |
| Software, algorithm | Adobe Illustrator 2025 | Adobe | RRID:SCR_010279 | |
| Software, algorithm | Adobe Premiere Pro 2024 | Adobe | RRID:SCR_021315 | |

## Zebrafish husbandry

Zebrafish work was conducted in compliance with animal ethics committees at the Peter MacCallum Cancer Centre, The University of Melbourne and The University of Queensland. The following previously published transgenic lines were used in this study: *TgBAC(pdgfrb:EGFP)*[uq15bh] (*Bower et al., 2017*), *Tg(kdrl:Hsa.HRAS-mCherry)*[s916] (*Hogan et al., 2009*), *Tg(kdrl:EGFP)*[s843] (*Jin et al., 2005*), *Tg(gata1:DsRed)*[sd2] (*Traver et al., 2003*), *TgBAC(acta2:EGFP)*[uq17bh] (*Bower et al., 2017*) , and *Tg(5xUAS:RFP)*[nkuasrfp1a] (*Asakawa et al., 2008*).

To ensure optical transparency, live imaging was performed in *slc45a2* (ENSDARG00000002593) F0 knockout animals as previously described (*Davis et al., 2021*) or by adding 0.003% 1-phenyl-2-thiourea (PTU) to embryo water at 1 day post-fertilization (dpf).

## BAC recombineering and transgenesis

The *TgBAC(abcc9:abcc9-T2A-mCherry)*[uom139] transgenic line was generated by BAC recombineering as previously described (*Suster et al., 2009*; *Bussmann and Schulte-Merker, 2011*). Briefly, T2A-mCherry insert with a kanamycin cassette was amplified by PCR using *pCS2-T2A-mCherry-KanR* plasmid as template. The primers used for amplifying the insert were:

5′-atggagcaggaggacggcctgtttgcatcttttgtcaaagccgacatgGAGGGCAGAGGAAGTCTGCTA-3′, 5′-aaaatggctttattgatctgttaaggccaaaagtggtgtaaagtggggaTCAGAAGAACTCGTCAAGAAGGCG-3′ (lowercase letters indicate homology arms to the BAC vector; uppercase letters indicate primer-binding sites on the template plasmid).

The amplified insert was introduced into *CH211-58C15* (*abcc9*) BAC clone containing a pRedET plasmid (GeneBridge, Heidelberg, Germany) and iTol2 cassette. Approximately 1 nl of purified BAC DNAs (100 ng/µl) mixed with *tol2* transposase mRNA (25 ng/µl) were injected into single-cell stage embryos. Injected fish were raised to adulthood and screened by fluorescence and PCR for germline transmission. The *pdgfrb:gal4FF* BAC construct was generated previously (*Ando et al., 2016*) and was injected to produce the *TgBAC(pdgfrb:gal4FF)*[uom140] transgenic line. The *pdgfrb:gal4FF* BAC

construct was kindly provided by Dr. Nathan Lawson (Department of Molecular, Cell, and Cancer Biology, University of Massachusetts Chan Medical School, USA).

## Genome editing and genotyping

The *pdgfrb*<sup>uq30bh</sup> mutant strain was generated using CRISPR/Cas9-mediated genome editing. The resulting allele carries a 39-bp deletion and a 5-bp insertion in exon 3 of *pdgfrb* (ENSDARG00000100897), leading to a predicted frameshift and premature stop codon.

> The gRNA sequence and genotyping primers used were: gRNA-binding site: 5′-GATGGTGACTAAGACGCGA-3′
> Forward genotyping primer: 5′-CTTCCTTAGATCCTGACGTGTG-3′
> Reverse genotyping primer: 5′-TATTGATGGGTTCGTCACCAG-3′

F0 *pdgfrb* crispants used were generated using Alt-R CRISPR–Cas9 reagents (IDT). The gRNA sequence and genotyping primers used were:

> Dr.Cas9.PDGFRB.1.AA: 5′-GATGGTGACTAAGACGCGAG-3′
> Forward genotyping primer: 5′-CTTCCTTAGATCCTGACGTGTG-3′
> Reverse genotyping primer: 5′-TATTGATGGGTTCGTCACCAG-3′
> Dr.Cas9.PDGFRB.1.AB: 5′-CTCGGTGCACACATAAACCC-3′
> Forward genotyping primer: 5′-GACGAGAACATCCCAGACTTTC 3′
> Reverse genotyping primer: 5′-GCGTGTAAACAAATCCTAACGG-3′

Throughout this study, sibling control fish were pooled homozygous wild-type and heterozygous animals, as no significant differences were detected between these genotypes (e.g. *Figure 1—figure supplement 1*).

## Fluorescent tracer injections

Fluorescently conjugated tracers used were: 1-kDa NHS Ester–Alexa Fluor 405 (Thermo Fisher: A30000), 10-kDa Dextran–Cascade Blue (Thermo Fisher: D1976), 10-kDa Dextran–Alexa Fluor 647 (Thermo Fisher: D22914), 70-kDa Dextran–Fluorescein (Thermo Fisher: D1822), and 2000-kDa Dextran–Tetramethylrhodamine (Thermo Fisher: D7139) each at 5 mg/ml.

Larvae (3–7 dpf) were anesthetized with tricaine and embedded in 0.5% low-melting agarose (Bio-Rad 1613112) in E3 water on 35 mm glass-bottom dishes (MatTek P35G-1.5-20-C). Approximately 5 nl was injected into the posterior cardinal vein (PCV). At 14 dpf, tracers were injected into the PCV with ~10 nl tracer on 3% agarose pads. In juveniles and adults retro-orbital injections were performed (*Pugach et al., 2009*) on 3% agarose pads, with 0.2 or 0.5 µl tracer, respectively. Tracers were injected individually or co-injected (1–70 kDa with 2000 kDa) depending on experimental design.

## Imaging

### Sample preparation

For live imaging at 3–14 dpf, larvae were mounted in 0.5% low-melting point agarose in E3 embryo water on a 35-mm glass bottom dish as previously described (*Okuda et al., 1846*). For fixed tissue imaging, zebrafish were euthanized with tricaine overdose (10 g/l), fixed (as whole or after brain extraction) in 4% paraformaldehyde at 4°C overnight and washed in PBS. Brains were embedded in 7% low-melting agarose in PBS (Tissue-Tek cryomolds) and coronally sectioned at 200 µm thickness using a vibrating microtome (Leica VT1000 S). Sections were mounted in RapiClear 1.52 solution (SunJin Lab, RC152201) for 3 hr at room temperature prior to imaging. Whole fish or brain samples were cleared using CUBIC clearing adapted from a previously described protocol (*Matsumoto et al., 2019*). Briefly, samples were incubated in CUBIC-L solution at 37°C overnight after fixation, washed and incubated in PBS at 4°C overnight, then incubated in CUBIC-R solution at room temperature overnight. Samples were embedded in 2% agarose in CUBIC-R on a 35-mm glass bottom dish for imaging.

## Image acquisition

Imaging was performed using the following microscopes and objectives: Olympus FV-MPERS upright multiphoton, 25x water dipping 1.05 NA (*Figure 1a, b, f*; *Figure 5—figure supplement 1e*), Olympus FV3000 confocal 10x air 0.4 NA (*Figures 3a–a'*, *4b–b'*, *c'*, *d–d'*, *and 5a–a'*; *Figure 5—figure supplement 1f*) and 40x oil immersion 1.4 NA (*Figure 5d, e*), Olympus/Evident FV4000 confocal 10x air 0.4 NA (*Figure 2—figure supplement 1d, e*; *Figure 3b–b'*; *Figure 3—figure supplement 1b, d*; *Figure 5b, f, g*; *Figure 5—figure supplement 2a–b"*) and 30x silicone immersion 1.05 NA (*Figure 1—figure supplement 1b*; *Figure 2a, c, e*; *Figure 2—figure supplement 1b*; *Figure 3c*; *Figure 3—figure supplement 1a–a"*, *b'*, *c-c'*; *Figure 5b'*) Nikon TiE with Yokogawa CSU-W1 spinning disk, Andor Sona sCMOS camera, 10x air 0.45 NA (*Figure 5—figure supplement 2a, b*), 20x air 0.75 NA (*Figure 2—figure supplement 2c*; *Figure 5—figure supplement 1a'*) and 40x water immersion 1.15 NA (*Figure 2—figure supplement 2a*). For tracer leakage assays, live imaging was performed at 2 hours post-injection (hpi), with a maximum variation in timing of 15 min due to the sequential imaging. Fixed tissue samples were collected at 0.5 hpi (*Figure 5f*, *Figure 5—figure supplement 2*), 2 hpi (*Figure 2—figure supplement 1d, e*, *Figures 4b–d'* and *5a–b'*,*d, e*, *Figure 5—figure supplement 1a, b*) or 6 hpi (*Figure 5g*). Stereomicroscope images were acquired using an NSZ-606 Zoom Stereomicroscope with a Tucsen Photonics Co. camera using TCapture/IS Capture software.

## Transmission electron microscopy

Brains were extracted from euthanized 5-month-old zebrafish, fixed in 2.5% glutaraldehyde for 1 hr at room temperature and washed in PBS. For sectioning, the brains were embedded in 7% low-melting point agarose in PBS in cryomolds (Tissue-Tek) and were sectioned coronally using a vibrating microtome (Leica VT1000 S) with 200 μm thickness. Tissue sections were processed as described previously (*Lim et al., 2017*). Briefly, tissue sections were immersed consecutively in a series of aqueous solutions: 1.5% potassium ferricyanide and 2% osmium tetroxide, 1% thiocarbohydrazide, 2% osmium tetroxide, 2% uranyl acetate and 0.06% lead nitrate using a Pelco Biowave at 80 W under vacuum for 3 min each. Vibratome sections then underwent serial dehydration in increasing concentrations of ethanol, before serial infiltration with increasing concentrations of Procure 812 resin. Ultrathin sections were obtained using a Leica UC64 ultramicrotome and imaged on a JEOL JEM-1011 at 80 kV.

## Image processing

Images were processed using Fiji (*Schindelin et al., 2012*) or Imaris 10.1 (Bitplane). Fluorescent images are shown as maximum intensity projections (MIPs), unless otherwise indicated. Vascular tracing (*Figure 1c*) and masked head regions (*Figure 1—figure supplement 1b*) are presented as snapshots, and videos highlighting the hotspot leakage sites were generated using Imaris 10.1 and annotated in Adobe Premier Pro 2024. Schematics were prepared in Adobe Illustrator 2025, and pseudocolor overlays on transmission electron microscopy images were generated in Adobe Photoshop 2024.

## Quantification of phenotypes

### Gross anatomy

Prior to imaging and genotyping, fish body lengths were measured from the tip of the head to the tail base. Body length datasets were compiled from multiple independent experiments. Survival was quantified using separately housed *pdgfrb*^*uq30bh*^ mutants and siblings to enable continuous tracking. Sibling and mutant groups were initially sorted based on *pdgfrb:EGFP*-positive cells in the brain using *TgBAC(pdgfrb:EGFP)*^*uq15bh*^, and were genotyped by PCR at the endpoint for each animal. Cerebrovascular aneurysms in juvenile *pdgfrb* mutants were qualitatively classified as mild or severe on the basis of aneurysm size relative to sibling controls.

## Mural cell number, vessel length, diameter, and branching points

Mural cell number was quantified by counting *pdgfrb:EGFP*-positive cells (*TgBAC(pdgfrb:EGFP)*^*uq15bh*^) in the brain associated with blood vessels (*Tg(kdrl:Hsa.HRAS-mCherry)*^*s916*^) in 3D view using Imaris 10.1. Vessel length, diameter, and branching points were quantified in Imaris 10.1. At larval stages, the lumenized midbrain central arteries (*Tg(kdrl:Hsa.HRAS-mCherry)*^*s916*^) branching from the middle mesencephalic central arteries (MMCtA) was traced using the Filament tool and the total filament length (μm) was plotted. Branching points within the traced vasculature were counted manually in 3D

view. For adults, 100-µm-thick MIPs were generated from whole brain imaging, using the continuation of the left MMCtA as an anatomical landmark to define a consistent region of interest. For diameter measurements, 10 capillary diameters were measured per sample. Vessel anatomy was assigned using (*Isogai et al., 2001*) as reference.

### Tracer intensity

The quantification of tracer intensity was performed on genotype blinded image sets using Imaris 10.1 (Bitplane) and Fiji (*Schindelin et al., 2012*) in 50-µm-thick MIPs, generated starting from 50 µm below the dorsal longitudinal vessel (DLV) to avoid including potential leakage from surface vessels. For live imaging quantifications, two different methods were used to validating findings. In the first method, the midbrain region was masked using the tracer accumulation in the skin as an outer boundary and metencephalic vessels as the border with the hindbrain. Vasculature was then masked and removed using *Tg(kdrl:EGFP)ˢ⁸⁴³* or *Tg(kdrl:Hsa.HRAS-mCherry)ˢ⁹¹⁶*, and mean intensity of extravasated tracer within the midbrain was measured. Relative leakage by molecular weight in wild-type animals was assessed by normalizing extravasated midbrain tracer intensity to intravascular tracer intensity. For comparisons between groups (siblings vs *pdgfrb* mutants, or for wild-type at 3 vs 7 dpf), extravasated midbrain tracer intensity was normalized to vascular reporter intensity (*kdrl:EGFP* or *kdrl:Hsa.HRAS-mCherry*). In the second method, in experiments including 2000-kDa dextran as an injection control, extravasated 70- or 10-kDa dextran intensity in midbrain parenchyma was calculated from the average intensity in four defined parenchymal regions. These values were normalized to the vascular intensity of 2000-kDa dextran, calculated from the average intensity in four regions within the DLV lumen. As such, for each data point: intensity = parenchymal tracer (10 or 70 kDa) intensity/luminal 2000-kDa tracer intensity. For datasets assessing leakage between siblings vs *pdgfrb* mutants at 7 and 14 dpf (*Figure 2*, *Figure 2—figure supplement 2*), we assessed the raw (non-normalized) tracer intensity in the parenchyma and compared these values to normalized values. We found that both normalization strategies improved the analysis, reducing variance and controlling for differences in tracer delivery and image variation between animals.

For tracer intensity quantification in sectioned brains, 50-µm-thick MIPs were generated. Tracer intensity outside the vasculature was measured at ~70-µm intervals from medial to lateral within a 0.7-mm region in juveniles, and at ~100-µm intervals within a 1-mm region in adults, due to differences in brain size. For tracer intensity quantification in sectioned brains, 50-µm-thick MIPs were generated and dextran intensity outside the vessels was measured at approximately every 70 µm from medial to lateral direction within 0.7 mm in juveniles and at every 100 µm within 1 mm in adults due to size difference. These measurements were normalized to vascular reporter intensity (*kdrl:EGFP*). MMCtAs served as anatomical landmarks to ensure consistent region selection.

### Hotspot leakage sites

Hotspot leakage sites were identified as prominent focal accumulations of fluorescent tracer adjacent to and outside the vasculature. Quantification was performed in 3D using Imaris 10.1.

### TEM quantification

Imaged sections were quantified using Fiji[66]. Caveolae were defined as circular profiles of less than 100 nm in diameter and were scored as luminal or abluminal based on proximity to each surface membrane (within 500 nm of each surface or in a thin-walled vessel the caveolae closest to each surface). Basement membrane thickness was scored in six different locations per vessel that were selected at random.

### Statistical analysis

Graphic representations and statistical analyses were performed using GraphPad Prism (*Hellström et al., 1999*). Shapiro–Wilk test was applied to test normal distribution of the data. Two-tailed Student's *t*-tests were used for two-group comparisons with normal distribution, and Mann–Whitney *U* tests for non-normal data. One- or two-way ANOVA followed by Tukey's post hoc test was used for multiple group comparisons. $p < 0.05$ was considered statistically significant and no data was excluded from the statistical analysis. Sample sizes were determined according to the general standards in the field.

## Materials availability

All materials used in the study are available upon request.

## Acknowledgements

This project was supported in part by funding from BrightFocus (A2018807S), the Australian Research Council (ARC) (DP210102712) and The Brain Cancer Centre (founded by Carrie's Beanies for Brain Cancer). BMH was supported by a National Health and Medical Research Council Senior Research Fellowship (1155221) and Investigator grant (2033008). RGP was supported by an ARC Laureate Fellowship (FL210100107). We thank the Centre for Advanced Histology and Microscopy (RRID:SCR_025432) at the Peter MacCallum Cancer Centre and Microscopy Australia Research Facility at the Centre for Microscopy and Microanalysis at The University of Queensland. We thank Koji Ando and Nathan Lawson for providing constructs and protocols for BAC recombineering. We also thank Kelly Smith and Marcos Sande Melon for academic discussions and technical suggestions.

## Additional information

### Competing interests

Benjamin M Hogan: Reviewing editor, eLife. The other authors declare that no competing interests exist.

### Funding

| Funder | Grant reference number | Author |
| --- | --- | --- |
| Australian Research Council | DP210102712 | Benjamin M Hogan |
| National Health and Medical Research Council | 1155221 | Benjamin M Hogan |
| National Health and Medical Research Council | 2033008 | Benjamin M Hogan |
| Australian Research Council | FL210100107 | Robert G Parton |
| BrightFocus Foundation | A2018807S | Benjamin M Hogan |

The funders had no role in study design, data collection and interpretation, or the decision to submit the work for publication.

### Author contributions

Oguzhan F Baltaci, Conceptualization, Data curation, Formal analysis, Validation, Visualization, Methodology, Writing – original draft, Writing – review and editing; Andrea Usseglio Gaudi, Stefanie Dudczig, Weili Wang, Scott Paterson, Maria Cristina Rondon-Galeano, Ye-Wheen Lim, James Rae, Alison Farley, Investigation, Methodology; Anne Lagendijk, Conceptualization, Supervision; Robert G Parton, Formal analysis, Investigation, Methodology; Benjamin M Hogan, Conceptualization, Formal analysis, Supervision, Funding acquisition, Writing – original draft, Writing – review and editing

### Author ORCIDs

Oguzhan F Baltaci ⬤ https://orcid.org/0009-0001-5651-1331
Robert G Parton ⬤ https://orcid.org/0000-0002-7494-5248
Benjamin M Hogan ⬤ https://orcid.org/0000-0002-0651-7065

### Ethics

Zebrafish work was conducted in compliance with animal ethics committees at the Peter MacCallum Cancer Centre, The University of Melbourne and The University of Queensland.

Reviewer #1 (Public review): https://doi.org/10.7554/eLife.104061.3.sa1

Reviewer #2 (Public review): https://doi.org/10.7554/eLife.104061.3.sa2
Author response https://doi.org/10.7554/eLife.104061.3.sa3

## Additional files

### Supplementary files
MDAR checklist

Source data 1. Numerical data used to generate all plots presented in the figures. Source data in an excel spreadsheet in which each separate tab provides the individual measurements and numbers used to generate the plots presented in each figure.

### Data availability
*Source data 1* contains the numerical data used to generate the figures.

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
