## [Editor Report · eLife Assessment]

This **important** study addresses the contribution of pericytes to the organization and permeability control of the zebrafish blood-brain barrier (BBB). By analyzing pdgfrb mutant zebrafish that lack brain pericytes, the authors reveal that the resulting cerebrovascular network is abnormally patterned. Remarkably, however, the barrier retains its restrictive permeability during larval and juvenile stages. More pronounced vascular defects become evident in adults, where localized BBB leakage coincides with hemorrhages and aneurysm formation. Based on **convincing** and beautifully documented imaging data, the authors argue that, unlike what has been reported in rodent systems, pdgfrb-dependent pericytes are not essential for maintaining BBB integrity in the zebrafish brain.

---

## [Referee Report · Reviewer #1 (Public review)]

Summary:

The study investigates the role of vascular mural cells, specifically pericytes and vascular smooth muscle cells (vSMCs), in maintaining blood-brain barrier (BBB) integrity and regulating vascular patterning. Analyzing zebrafish pdgfrb mutants that lack brain pericytes and vSMCs, the show that mural cell deficiency does not impair BBB establishment or maintenance during larval and early juvenile stages. However mural cells seem to be crucial for preventing vascular aneurysms and hemorrhage in adulthood as focal leakage, basement membrane disruption and increased caveolae formation are observed in adult zebrafish at aneurysm hotspots. The authors challenge the paradigm that mural cells are essential for BBB regulation in early development while highlighting their importance for long-term vascular stability.

Strengths:

Previous studies have established that the zebrafish BBB shares molecular and morphological homology with e.g. the mammalian BBB and therefore represents a suitable model. By examining mural cell roles across different life stages-from larval to adult zebrafish-the study provides an unprecedented comprehensive developmental analysis of brain vascular development and of how mural cells influence BBB integrity and vascular stability over time. The use of live imaging, whole-brain clearing, and electron microscopy offers high-resolution insights into cerebrovascular patterning, aneurysm development, and structural changes in endothelial cells and basement membranes. By analyzing "leakage hotspots" and their association with structural endothelial defects in adults the presented findings add novel insights into how mural cell loss may lead to vascular instability.

---

## [Referee Report · Reviewer #2 (Public review)]

Summary:

The authors generated a zebrafish mutant of the pdgfrb gene. The presented analyses and data confirm previous studies demonstrating that Pdgfrb signaling is necessary for mural cell development in zebrafish. In addition, the data support previously published studies in zebrafish showing that mural cell deficiency leads to hemorrhages later in life. The authors presented quantified data on vessel density and branching, assessed tracer extravasation, and investigated the vasculature of adult mice using electron microscopy.

Strengths:

The strength of this article is that it provides independent confirmation of the important role of Pdgfrb signaling for the development of mural cells in the zebrafish brain. In addition, it confirms previous literature on zebrafish that provides evidence that, in the absence of pericytes/VSMC, hemorrhages appear (Wang et al, 2014, PMID:24306108 and Ando et al 2021, PMID:3431092)".

The Reviewing Editor has carefully reviewed the revised manuscript and is fully satisfied with the authors' revisions.

---

## [Author Response]

The following is the authors’ response to the original reviews.

**Public Reviews:**

**Reviewer #1 (Public review):**
Summary:The study investigates the role of vascular mural cells, specifically pericytes and vascular smooth muscle cells (vSMCs), in maintaining blood-brain barrier (BBB) integrity and regulating vascular patterning. Analyzing zebrafish pdgfrb mutants that lack brain pericytes and vSMCs, they show that mural cell deficiency does not impair BBB establishment or maintenance during larval and early juvenile stages. However, mural cells seem to be crucial for preventing vascular aneurysms and hemorrhage in adulthood as focal leakage, basement membrane disruption, and increased caveolae formation are observed in adult zebrafish at aneurysm hotspots. The authors challenge the paradigm that mural cells are essential for BBB regulation in early development while highlighting their importance for long-term vascular stability.Strengths:Previous studies have established that the zebrafish BBB shares molecular and morphological homology with e.g. the mammalian BBB and therefore represents a suitable model. By examining mural cell roles across different life stages - from larval to adult zebrafish - the study provides an unprecedented comprehensive developmental analysis of brain vascular development and of how mural cells influence BBB integrity and vascular stability over time. The use of live imaging, whole-brain clearing, and electron microscopy offers high-resolution insights into cerebrovascular patterning, aneurysm development, and structural changes in endothelial cells and basement membranes. By analyzing "leakage hotspots" and their association with structural endothelial defects in adults the presented findings add novel insights into how mural cell loss may lead to vascular instability.Weaknesses:The study uses quantitative tracer assays with multiple molecular weight dyes to evaluate blood-brain barrier (BBB) permeability. The study normalizes the intensity of tracer signals (e.g., 10 kDa, 70 kDa dextrans) in the brain parenchyma to the vascular signal of a 2000 kDa dextran tracer (assumed to remain within vessels). Intensity normalization is used to control for variations in tracer injection efficiency or vascular density. This method doesn't directly assess the absolute amount of tracer present in the parenchyma, potentially underestimating leakage severity. As the lack of BBB impairment is a "negative" finding, more rigorous controls or other methods might be needed to corroborate it.

In response to these and comments from other reviewers, we have now performed further carefully controlled analysis to test leakage of tracers using molecular weights ranging from 1 to 2000 kDa. We have performed additional normalisation approaches (new data in Fig. 2a–d) imaging tracer extravasation together with vascular reporters (*Tg*(*kdrl:EGFP*)*^s843^* or *Tg*(*kdrl:Hsa.HRAS-mCherry*)*^s916^*) and used this transgenic reporter for normalisation (as suggested by Reviewer #2). The results of these experiments all supported our initial conclusions (revised Extended Data Fig. 3a–d) further validating the reliability of our method. Furthermore, as suggested by the reviewer analysis of the raw tracer intensity amounts in the parenchyma were also performed with no normalization at all (see Author response image 1). This also supports our conclusion that the BBB is intact in young animals. Finally, we now use our methods to demonstrate that we can detect an immature leaky BBB at 3 dpf and a mature functional BBB at 7 dpf (Fig. 2e-f), a suitable positive control to show that our methods and analyses are reliable.

**Author response image 1. sa3fig1:** Raw intensity values from the parenchyma confirm findings in Figure 2 and Extended Data Figure 3. a–d, Raw mean fluorescence intensity values of extravasated tracers in the midbrain.(a–b) show unnormalized values corresponding to Extended Data Fig. 3a–d, and (c–d) show unnormalized values corresponding to Fig. 1a–d. Unpaired t-tests for 70 and 10 kDa at 14 dpf in (a–b), for 10 kD at 7 dpf, and for 70 kDa at 14 dpf in (c–d). Mann-Whitney tests for 70 and 10 kDa at 7 dpf in (a–b), for 70 kDa at 7 dpf, and for 10 kDa at 14 dpf (c–d), due to non-normal distribution. These data were all generated in genotype blind assays, display variance in signal that is generated between embryos due to injection differences and show no difference between the genotypes analyzed in BBB integrity. Comparison of this to normalised data using 2000 kDa tracer or *kdrl* expression in endothelial cells (Fig. 2 and Extended Data Fig. 3) confirms that normalisation improves the analysis, effectively controlling for embryo-to-embryo differences in delivery of tracer and imaging.

**Reviewer #2 (Public review):**
Summary:The authors generated a zebrafish mutant of the pdgfrb gene. The presented analyses and data confirm previous studies demonstrating that Pdgfrb signaling is necessary for mural cell development in zebrafish. In addition, the data support previously published studies in zebrafish showing that mural cell deficiency leads to hemorrhages later in life. The authors presented quantified data on vessel density and branching, assessed tracer extravasation, and investigated the vasculature of adult mice using electron microscopy.Strengths:The strength of this article is that it provides independent confirmation of the important role of Pdgfrb signaling for the development of mural cells in the zebrafish brain. In addition, it confirms previous literature on zebrafish that provides evidence that, in the absence of pericytes/VSMC, hemorrhages appear (Wang et al, 2014, PMID: 24306108 and Ando et al 2021, PMID: 3431092). The study by Ando et al, 2021 did not report experiments assessing BBB leakage in pdgfrb mutants but in the review article by Ando et al (PMID: 34685412) it is stated that "indicating that endothelial cells can produce basic barrier integrity without pericytes in zebrafish."

We thank the reviewer for their comments and pointing out literature that we had not cited (this has been corrected in our revised manuscript).

As noted by other reviewers, our study goes beyond simply confirming previous literature. The quoted section by the reviewer from Ando et al 2021 regarding intact barrier integrity in *pdgfrb* mutants is a conclusion based on apparent lack of haemorrhages in *pdgfrb* mutants[1]. Our work shows haemorrhages in older animals and as such is in line with these previously published results, but it also extends previous work, for the first time reporting detailed functional analysis to assess BBB integrity. Our study uses definitive tracer assays (now including extensive revisions) to identify intact the BBB in *pdgfrb* mutants in live animals. This has not been previously described and is important because it offers a new perspective on the evolutionary conservation (or otherwise) of pericyte control of BBB function. Furthermore, our study investigates the nature of hotspot leakage and haemorrhages in more detail than in previous work.

Weaknesses:(1) The authors should avoid using violin plots, which show distribution. Instead, they should replace all violin plots in the figures with graphs showing individual data points and standard deviation. For Figure 2f specifically, the standard deviation in the analyzed cohort should be shown.

This is a good point and we have replaced the violin plots with individual data points and shown all data as mean ± SEM.

(2) The authors have not shown the reduced PDGFRB protein or the effect of mutation on mRNA level in their zebrafish mutant.

Our *pdgfrb^uq30bh^* mutant allele introduces a mutation predicted to generate a truncated protein very similar to previously validated alleles (see detail in revised Extended Data Fig. 1a and methods). Our *pdgfrb^uq30bh^* mutant also phenocopies previous *pdgfrb* mutants (*sa16389* and *um148* alleles)[2,3], displaying mural cell loss with multiple markers (Fig. 1a, new data in Extended Data Fig. 1b–c, Fig. 3b–c; Extended Data Fig. 4c–d) and the same typical morphological defects and survival rates (new data in Extended Data Fig. 1d–f). Thus our mutant phenocopy gives confidence it is most likely a null allele, in line with previous papers studying presumed null alleles[1].

We believe this provides sufficient confidence in this allele of *pdgfrb.* Moreover, considering that our manuscript focusses on loss of mural cells and we show definitively that this mutant has robust loss of mural cells in the brain, our mutant is suitable for this study.

(3) Statistical data analysis: Did the authors perform analyses to investigate whether the data has a normal distribution (e.g., Figures 1d, e)?

We thank the reviewer for raising this and apologise for this oversight. All data have now been assessed for normality using Shapiro-Wilk test and further statistical analyses have been performed accordingly. The specific quantifications referred to by the reviewer in Extended Data Fig. 3a–d (previously Fig. 1d-e), have normal distribution except for quantification measuring 70 kDa extravasation at 7 dpf, therefore Mann-Whitney test has been used for this comparison. Further information can be found in figure legends and methods.

(4) Analysis of tracer extravasation. The use of 2000 kDa dextran intensity as an internal reference is problematic because the authors have not provided data demonstrating that the 2000 kDa dextran signal remains consistent across the entire vasculature. The authors have not provided data demonstrating that the 2000 kDa dextran signal in vessels exhibits acceptable variance across the vasculature to serve as a reliable internal reference. The variability of this signal within a single animal remains unknown. The presented data do not address this aspect.

We thank the reviewer for their comment and agree that analysis was needed for showing 2000 kDa dextran as a reliable normalization signal.

We now show the data in the following Figures that demonstrate the consistency of signal throughout the vasculature using this 2000-kDa tracer: Extended Data Fig. 2b, Extended Data Fig. 3a and c, Extended Data Fig. 5a, Extended Data Fig. 6. In fact, we observe that this 2000 kDa tracer provides a very reliable marker of large and small calibre vessels in larval, juvenile and adult animals, even in fixed and cleared whole tissues and animals (e.g. Extended Data Fig. 2d-e, Extended Data Fig. 5 and 6).

Our further experiments and analysis support the use of this tracer as an ideal way to normalise for variation between animals and coupled with improved masking of vessels using transgenic labels (e.g. Extended Data Fig. 2b) we can quantify across whole vascular networks to reduce the concern about variation within individual animals. We also find 2000 kDa shows negligible leakage through the brain vessels Extended Data Fig. 2b–c (new data) at 2 hours post-injection (hpi) and provided images in Extended Data Fig. 6b–b′′ showing detectable signals even at 6 hpi. Finally, results generated with this approach, normalisation to transgenic markers or even raw parenchymal values of tracer intensity, generate the same conclusions. In addition, we point the reviewer to a recent pre-print that further validates this method from our team[4].

Overall, we find the use of this tracer an ideal way to normalise for differences in injection volumes between animals and we recommend the use of this method to other groups assessing BBB leakage in zebrafish.

Additionally, it's intriguing that the signal intensity in the parenchyma of the tested tracers presents a substantial range, varying by 20-30% in the analysed cohort (Figure 1g, Extended Figure 1e). Such large variability raises the question of its origin. Could it be a consequence of the normalization to 2000 kDa dextran intensity which differs between different fish? Or is it due to the differences in the parenchymal signal intensity while the baseline 2000 kDa intensity is stable? Or is the situation mixed?

This is a good point raised by the reviewer.

To address this, we have used the following approaches:

(1) We provide additional experiments and normalisation methods that support the utility of our tracer studies (new data in Fig 2a–f and Extended Data Fig. 2b–c), discussed in detail below.

(2) We provide graphs of the raw parenchymal distribution of tracer not normalised at all (also requested by reviewer 1). This is provided in Author response image 1 and further supports all our conclusions, showing that our normalisation methods generate meaningful data.

Overall, the range of parenchymal intensity that we see after tracer injection and live imaging shows variations introduced during microinjection. However, these ranges are in-line with previous publications using similar methods (see studies by O’Brown et al 2019 and 2023)[5,6], allow reliable statistical comparisons to be drawn between control and mutants and allow us to detect both immature and functional BBB states during zebrafish development (new data in Fig. 2e-f).

Of note, the variability we see is likely introduced during the injection process into tiny larval blood vessels and is the reason why we perform normalization of parenchymal tracers to a vascular dextran signal that doesn’t leak from brain vessels. In our studies, 2000-kDa dextran has been co-injected with the smaller size tracers, therefore any potential differences in injection volumes as well as imaging conditions (however consistent) should be reduced by this method.

An alternative and potentially more effective approach would be to cross the pdgfrb mutant line with a line where endothelial cells are genetically labeled to define vessels (e.g. the line kdrl used in acquiring data presented in Figure 2a). Non-injected controls could then be used as a baseline to assess tracer extravasation into the parenchyma.

We thank the reviewer for this suggestion.

In response, we have performed new tracer leakage experiments at 7 and 14 dpf in siblings and pdgfrb mutants and quantified parenchymal tracer extravasation by normalizing to vascular reporters (*Tg*(*kdrl:EGFP*)*^s843^* or *Tg*(*kdrl:Hsa.HRAS-mCherry*)*^s916^*). The results were in-line with the previously presented and independent experiments and showed indistinguishable phenotypes between siblings and pdgfrb mutants (new data, Fig. 2a–d). We also used uninjected controls to assess baseline and saw consistent values approaching zero in these images and did not include this in the revised paper.

Furthermore, we have also used this approach in wild-type larvae at 3 dpf (immature BBB) and 7 dpf (functional BBB)[5]. We detected significantly higher parenchymal extravasation of 10 and 70 kDa tracers at 3 dpf compared to 7dpf, demonstrating that our method can detect leakage (new data, Fig. 2e–f).

We believe that both normalization approaches have advantages (as discussed above), therefore showing the same results with these two different approaches has further strengthened our findings.

How is the data presented in Figure 3e generated? How was the dextran intensity calculated? It looks like the authors have used the kdrl line to define vessels. Was the 2000 kDa still used as in previous figures? If not, please describe this in the Materials and Methods section.

We have moved this data to Fig. 4e (previously Fig. 3e).

Previously, we had plotted raw data due to the nature of the experiment being conducted on a vibratome sectioned tissue. The 2000 kDa tracer was not used. In response to this query and to be consistent with the new approach suggested by the reviewer, we have revised the quantification by normalizing the 10 kDa tracer extravasation to *Tg*(*kdrl:Hsa.HRAS-mCherry*)*^s916^* for this and the new experiments on juveniles (Fig. 5h–i). Please see the corresponding figure legends or revised methods (lines 464–472).

(5) The authors state that both controls and mutants show extravasation of 1 kDa NHS-ester into the parenchyma. However, the presented images do not illustrate this; it is not obvious from these images (Extended Data Figure 1c). Additionally, the presented quantification data (Extended Data Figure 1e) do not show that, at 7 dpf, the vasculature is permeable to this tracer. Note that the range of signal intensity of the 1 kDa NHS-ester is similar to the 70 kDa dextran (Figure 1g and Extended Figure 1e). Would one expect an increase in the ratio in case of extravasation, considering that the 2000 kDa dextran has the same intensity in all experiments? Please explain.

We thank the reviewer for raising this important point.

To clarify, we have never claimed that “2000-kDa dextran has the same intensity in all experiments”. On the contrary, vascular 2000 kDa normalization has been used to account for potential differences caused by injection, as stated in the submitted supplementary materials and now made more clear in the revision.

In response to this query, we conducted more detailed analysis on tracer extravasation patterns based on molecular weight (new data, Extended Data Fig 2b–c). This analysis showed that 1- and 10-kDa tracers have much higher extravasation rate compared to 70- and 2000-kDa tracers. Interestingly, we did not find a significant difference between 1 and 10 kDa extravasation. Therefore, in the revised manuscript we used only 10 kDa in further experiments and have removed 1 kDa from the figures.

To assess the tracers individually (new data in Extended Data Fig. 2c), parenchymal extravasation of individual tracers was normalised to their own vascular signal (eg. Mean intensity of 10 kDa in midbrain/mean intensity of 10 kDa in vasculature), to account for potential differences in injection volume. This provides a suitable method to assess leakage in wild-type animals and is now in line with how previous studies have analysed such tracer injections[5,6]. Please see revised figure legends and supplementary materials for details.

(6) The study would be strengthened by a more detailed temporal analysis of the phenotype. When do the aneurysms appear? Is there an additional loss of VSMC?

We thank the reviewer for this suggestion, and we have now performed staged imaging of the pdgfrb mutants and siblings between 7 and 21 dpf using *TgBAC*(*acta2:EGFP*)*^uq17bh^* transgene (new data, Fig. 3b-c; Extended Data Fig. 4a–d). Consistent with previous results, *acta2*:*EGFP*-positive cells surrounding the middle mesencephalic central arteries (MMCtA) were missing in pdgfrb mutants. At 21 dpf, we have also observed a mild dilation of these vessels, likely the earliest changes to generate aneurysms (new data, Fig. 3c).

To extend the number of stages analysed in this study, we have also performed new tracer leakage experiments in juveniles (30 dpf) and found that aneurysms can be detected at this age when the 10 kDa tracer is used (new data in Fig. 5b–b′). Consistent with the adult stage phenotype, aneurysms were limited to the larger calibre vessels (arteries) in the brain. We have also observed hotspots, and upon quantification, we found fewer numbers in juveniles compared to adults, suggesting that severity of aneurysms and hotspots increase with age.

Taken together, our results show that the aneurysms in *pdgfrb* mutants start appearing at late larval/early juvenile stages (~21 dpf) with observable dilations. By 30 dpf, aneurysms accompanied by small numbers of hotspots are observed, which exhibits significantly increased numbers by adulthood. This also correlates with reduced development and survival rate of *pdgfrb* mutants after 30 dpf (new data, Extended Data Fig. 1d–e).

(7) The authors intended to analyze the BBB at later stages (line 128), but there is not a significant time difference between 2 months (Figure 2) and 3 months (Figure 3) considering that zebrafish live on average 3 years. Therefore, the selection of only two time-points, 2 and 3 months, to analyze BBB changes does not provide a comprehensive overview of temporal changes throughout the zebrafish's lifespan. How long do the pdgfb mutants live?

Respectfully, zebrafish transition from juvenile stages to adulthood between 2 and 3 months and there are many significant differences in the physiology of this organism at these two ages. At 2 months, zebrafish are still juveniles undergoing metamorphosis with rapid growth and ongoing skeletal and vascular development. By 3 months, they are sexually mature adults and have much more developed cranioskeletal and vascular systems. Having said that, we take the reviewers important point that further temporal resolution would improve the study.

We have performed new experiments in 1-month-old animals and provided comprehensive analysis of the vascular phenotypes occurring in *pdgfrb* mutants. These were very informative experiments analysing leakage using 10-kDa tracer injections and have significantly improved the study. We had previously provided experiments at 5-month-old adults as well (previously Fig. 4a–b and Extended Data Fig. 4a) and so now the study includes larval stages (7, 14 dpf), juveniles at 1 and 2 months and adults at 3 and 5 months. While the additional timepoints did not offer up any new conclusions, they significantly enhanced the body of work overall.

Of further note, we provided survival data up to 90 dpf where survival of the *pdgfrb* mutants is significantly reduced compared to siblings (Extended Data Fig. 1e). We believe this is associated with the severity of the aneurysms and haemorrhages which probably lead to lethality in these mutants.

(8) Why is there a difference in tracer permeability between 2 and 3 months (Figures 2 and 3)? Are hemorrhages not detected in 2-month-old zebrafish?

In response to this and other queries, we have added new additional experiments that provide more detailed temporal analysis on tracer accumulation (new data in Fig. 5b–c, Fig. 5f–g).

In short, we do not see obvious haemorrhages in 1- or 2-month fish at a gross level during dissections (not shown). We find that using 10-kDa tracer, we can detect small hotspots at aneurysms as early as 1 month, likely representing the earliest loss of integrity. We do not see obvious hotspots in 2-month-old animals when we use the 70-kDa tracer, this suggests to us that it is less sensitive for hotspot detection (in line with new Extended Data Fig. 2c). Finally, we find that the number of hotspots increases dramatically from Juvenile to Adult stages in our datasets, which we take as indicative of a progressive phenotype.

Overall, tracer size matters for detecting hotspots and they become more apparent in older animals - we have added a note in the main text to cover these points (lines 200–205)

(9) Figure 3: The capillary bed should be presented in magnified images as it is not clearly visible. Figure 3e shows that in the pdgfb mutant the dextran intensity is higher also in regions 6-10. How do the authors explain this?

We thank the reviewer for raising this important point.

Firstly, we now include enlarged views of the capillary beds for this experiment (Fig. 4d′) and new experiments mentioned below.

Secondly, in relation to why there is higher tracer in lateral locations and not just medial sites of haemorrhage, we believe that this is most likely due to the progressive spread of tracer from the medial hotspots. To test if this is likely, we performed additional experiments and tested tracer accumulation at 2 different timepoints in brains collected at 0.5 or 6 hpi (new data in Fig. 5f–g, Extended Data Fig. 6a–b′′). Tracer accumulation at 0.5 hpi was very minimal and was primarily limited to hotspots and nearby regions new data in (Fig. 5h), whereas a higher tracer accumulation in brains was observed across medial to lateral regions at 6 hpi (new data in Fig. 5i) in *pdgfrb* mutants. Comparing the data in Figure 4 (2 hpi) and new data in Figure 5i (6 hpi), the 10 kDa-tracer appears to have spread to more lateral locations given the increased time allowed post injection.

We cannot formally exclude the possibility that tracer leakage does occur slower through capillaries than at major hotspots, which might fit with the proposed model of slow leakage via increased EC transcytosis[7-9]. However, considering that we cannot detect increased tracer accumulation in *pdgfrb* mutants that lack aneurysms and haemorrhages at 7 and 14 dpf, such a scenario would require capillary transcytosis to be active at later juvenile and adult stages but not in larval and late larval animals. Thus, we believe the most plausible explanation is that aneurysm/haemorrhage associated leakage is the primary cause of the vascular integrity defects in zebrafish *pdgfrb* mutants.

We have added discussions addressing this in the revised manuscript (lines 220–230, 300–302).

(10) In general, the manuscript would benefit from a more detailed description of the performed experiments. How long did the tracer circulate in the experiments presented in Figures 2, 3, and 4?

We thank the reviewer for this suggestion and have now ensured that this is clearly described for in figure legends and methods (lines 391–395).

(11) How do the authors explain the poor signal of the 70 kDa dextran from the vasculature of 5-month-old zebrafish presented in Extended Data Figure 3?

We agree that the dextran signal was reduced compared to the other experiments in that Figure. This is likely due to sample preparation and clearing causing reduced fluorescence. Upon consideration of the presented data and the additional experiments using 10 kDa tracers providing further validations for our claims, we decided to remove this data from the paper.

(12) The study would benefit from a clear separation of the phenotypes caused by the loss of VSMC. The title eludes that also capillaries present hemorrhages which is not the case. How do vascular mural cells differ from mural cells? Are there any other mural cells?

We take the reviewers point and have now updated the title as "Mural cells protect the adult brain from haemorrhage but do not control the blood-brain barrier in developing zebrafish."

(13) I have a few comments about how the authors have interpreted the literature and why, in my opinion, they should revise their strong statements (e.g., the last sentence in the abstract).Scientists have their own insights and interpretations of data. However, when citing published data, it should be clearly indicated whether the statement is a direct quote from the original publication or an interpretation. In the current manuscript, the authors have not correctly cited the data presented in the two published papers (references 5 and 6). These papers do not propose a model where pericytes suppress "adsorptive transcytosis" (lines 73-76). While increased transcytosis is observed in pericyte-deficient mice, the specific type of vesicular transport that is increased or induced remains unknown.Similarly, lines 151-152 refer to references 5 and 6 and use the term "adsorptive transcytosis," but the authors of both papers did not use this term. Attributing this term to the original authors is inaccurate. Additionally, lines 152-153 do not accurately represent the findings of references 5 and 6. These papers do not state that there is an induction of "caveolae" in endothelial cells in pericyte-deficient mice. In the absence of pericytes, many vesicles can be observed in endothelial cells, but these vesicles are relatively large. It is more likely that there is some form of uncontrolled transcytosis, perhaps micropinocytosis. Please refer to the original papers accurately.

We thank the reviewer for these comments. We take the point and have rewritten the manuscript carefully to improve accuracy and avoid misrepresenting any previous claims made in specific papers.

Also, the authors have missed the fact that in mice, the extent of pericyte loss correlates with the extent of BBB leakage. To a certain extent, the remaining pericytes, can compensate for the loss by making longer processes and so ensure the full longitudinal coverage of the endothelium. This was shown in the initial work of Armulik et al (reference 5) and later in other studies.

We certainly did not miss this important point (as we are also working with these mouse models) and we now include reference to this in our expanded discussion. Of note, we do think it would be worthwhile assessing if the extent of BBB leakage and pericyte coverage also correlates with the presence of microhaemorrhages in these hypomorphic mouse models, although this is more challenging to do in mice than in zebrafish.

The bold assertion on lines 183 -187 that a lack of specific BBB phenotype in pdgfrb zebrafish mutant invalidates mouse model findings is unfounded. Despite the notion that zebrafish endothelium possesses a BBB, I present a few examples highlighting the differences in brain vascular development and why the authors' expectation of a straightforward extrapolation of mouse BBB phenotypes to zebrafish is untenable.In mice Pdgfrb knockout is lethal, but in zebrafish, this is not the case. In marked contrast to mice, however, zebrafish pdgfrb null mutants reach adulthood despite extensive cerebral vascular anomalies and hemorrhage. Following the authors' argumentation about the unlikely divergence of zebrafish and mice evolution, does it mean that the described mouse phenotype warrants a revisit and that the Pdgfrb knockout in mice perhaps is not lethal? Another example where the role of a gene product is not one-to-one, which relates to pericyte development, is Notch3. Notch3-null mice do not show significant changes in pericyte numbers or distribution, suggesting a less prominent role in pericyte development compared to zebrafish.Although many aspects of development are conserved between species, there are significant differences during brain vascular development between zebrafish and mice. These differences could reveal why the BBB is not impaired in zebrafish pdgfrb mutants. There is a difference in the temporal aspect when various cellular players emerge. The timing of microglia colonization in the brain differs. In mice, microglia colonization starts before the first vessel sprouts enter the brain, while in zebrafish, microglia enter after. Additionally, microglia in zebrafish and mice have a different ontogeny. In mice, astrocytes specialize postnatally and form astrocyte endfeet postnatally. In zebrafish, radial glia/astrocytes form at 48 hpf, and as early as 3 dpf, gfap+ cells have a close relationship with blood vessels. Thus, these radial glia/astrocyte-like cells could play an important role in BBB induction in zebrafish. It's worth noting that in Drosophila, the blood-brain barrier is located in glial cells. While speculative, these cells might still play a role in zebrafish, while the role of pericytes does not seem to be crucial. Pericytes enter the brain and contact with developing vasculature (endothelium) relatively late in zebrafish (60 hpf). In mice, the situation is different, as there is no such lag between endothelium and pericyte entry into the brain. I suggest that the authors approach the observed data with curiosity and ask: Why are these differences present? Are all aspects of the BBB induced by neural tissue in zebrafish? What is the contribution of microglia and astrocytes?"Another interesting aspect to consider is the endothelial-pericyte ratio and longitudinal coverage of pericytes in the zebrafish brain, and how this relates to what is observed in mice. How similar is the zebrafish vasculature to the mouse vasculature when it comes to the average length of pericytes in the zebrafish brain? Does the longitudinal coverage of pericytes in the zebrafish brain reach nearly 100%, as it does in mice?Based on the preceding arguments, it is recommended that the authors present a balanced discussion that provides insightful discussion and situates their work within a broader framework.

Overall, we agree with most of the points made by the reviewer above. As we have now extended the format of this paper to be a full article, we have space to provide an extended discussion and introduction. We now try to capture many of the points made by the reviewer and we think that this has significantly improved the paper. We thank the reviewer for this contribution.

We do want to point out that we did not state that our findings using zebrafish pdgfrb mutants invalidate mouse model findings. We suggest that a deeper analysis to understand the nature of the hotspots in mural cell deficient mammalian models could be very interesting in light of the zebrafish observations. We hope that the revised discussion better reflects this.

**Reviewer #3 (Public review):**
This manuscript examines the role of pdgfrb-positive pericytes in the establishment and maintenance of the blood-brain barrier (BBB) in the zebrafish. Previous studies in PDGFB- or PDGFRB-deficient mice have suggested that loss of pericytes results in disruption of the BBB. The authors show that zebrafish pdgfrb mutant larvae have an intact BBB and that pdgfrb mutant adult fish show large vessel defects and hemorrhage but do not exhibit substantial leakage from brain capillaries, suggesting loss of pericytes is not sufficient to "open" the BBB. The authors use beautiful and compelling images and rigorous quantification to back up most of their conclusions. The imaging of the adult brain is particularly nice. The authors rigorously document the lack of BBB leakage in pdgfrbuq30bh mutant larvae and large vessel phenotypes (eg, enlargement and rupture) in pdgfrbuq30bh mutant adults. A few points would help the authors to further strengthen their findings contradicting the current dogma from rodent models.

We appreciate the reviewer's comments on the manuscript overall and agree that addressing the raised points was needed to strengthen our findings. We have addressed the main points below and believe that this revision greatly improves this study.

Major point:The authors document pericyte loss using a single TgBAC(pdgfrb:egfp)ncv22 transgenic line driven by the promoter of the same gene mutated in their pdgfrbuq30bh mutants. Given their findings on the consequences of pericyte loss directly contradict current dogma from rodent studies, it would be useful to further validate the absence of brain pericytes in these mutants using one of several other transgenic lines marking pericytes currently available in the zebrafish. This could be done using pdgfrb crispants, which the authors show nicely phenocopy the germline mutants, at least in larvae. This would help nail down the absence of any currently identifiable pericyte population or sub-population in the loss of pdgfrb animals and substantially strengthen the authors' conclusions.

We thank the reviewer and agree that examination of *pdgfrb^uq30bh^* mutants using another transgenic line labelling pericytes would further validate the absence of brain pericytes. We generated a transgenic line, *TgBAC*(*abcc9:abcc9-T2A-mCherry*)*^uom139^*, to visualise pericytes and validated the absence of brain pericytes in the *pdgfrb* mutants (revised Extended Data Fig. 1b). The loss of brain pericytes matched our findings using *TgBAC*(*pdgfrb:egfp*)*^uq15bh^* line as well as previously published data by Ando et al 2016-2021, where the brain pericytes except for metencephalic artery were missing[2,3].

Other issues:The authors should provide more information about the pdgfrbuq30bh mutant and how it was generated (including a diagram in a supplemental figure would be useful).

We thank the reviewer for this suggestion. In addition to the explanations provided in supplementary materials, we have added a schematic, provided sanger sequencing results showing the mutation as well as predicted effect of the mutation on the protein domains (Extended Data Fig. 1a).

It would be helpful to show some data on whether mutants show morphological phenotypes or developmental delay at 7 and 14 dpf, to provide some context to better assess the reduced branching and vessel length vascular phenotypes (see Figures 1c-e).

We thank the reviewer for this suggestion. We have provided further details on body length and survival of the *pdgfrb* mutants until 90 dpf. As reported by Ando et al 2021, we did not observe any distinguishing feature until about 30 dpf[1,3]. The adult anatomy of our mutant allele matches that of previously described null mutants and is now shown (Extended Data Fig. 1f).

If available, it would be helpful to have a positive control for the tracer leakage experiments - a genetic manipulation that does cause disruption of the BBB and leakage at 2 hours post-tracer injection (see Figures 1f and g).

We thank the reviewer for this suggestion and agree that a positive control would validate reliability of our method. We have performed new experiments at 3 dpf when BBB integrity is not yet established and at 7 dpf when BBB is functional in zebrafish[5], testing both 10 and 70 kDa tracers (new data in Fig. 2e–f). We detected significantly higher tracer accumulation at 3 dpf, showing that our methods can detect tracer leakage in the brain.

Quantification of the findings in Figure 4c, d would be useful, as would the use of germline fish for these experiments if these are now available. If this is not possible, it would be helpful to document that the crispants used in these experiments lack pdgfrb:egfp pericytes at adult stages (this is only shown for 5 dpf larvae, in Extended Data Figure 4b).

We thank the reviewer for this comment. Using *TgBAC*(*pdgfrb:egfp*)*^uq15bh^* line, we have imaged coronal brain sections collected from 10-week old pdgfrb crispants and uninjected siblings (age-matched animals used in Fig. 5d–e, previously Fig. 4c–d). We have now included data showing that adult pdgfrb crispants lack brain mural cells, phenocopying *pdgfrb^uq30bh^* mutants (new data, Extended Data Fig. 6f). These particular crispants are very reliable in our hands and nicely reproduce stable mutant phenotypes, giving us confidence to use the faster F0 approach in this experiment.

Adult mutants clearly show less dye leakage in the more superficial capillary regions than WT siblings, but dextran intensity is a bit higher, although this could well be diffusion from more central brain regions where overt hemorrhage is occurring. Along similar lines though, the authors' TEM data in Extended Data Figure 4d hints that there may be more caveolae in mutant brain capillaries, although the N number was lower here than for the measurements from TEM of larger central vessels (Figure 4g). It would be useful to carry out additional measurements to increase the N number in Figure 4d to see whether the difference between wild-type sibling and mutant capillary caveolae numbers remains as not significant.

We thank the reviewer for these raising important points and suggestions.

Firstly, in relation to signal in capillary regions and likely diffusion from hotspots, please see the response to reviewer 3 point 9 above.

Secondly, we have imaged and analysed more capillaries in both *pdgfrb* mutants and siblings (Extended Data Fig. 7a–b, previously Extended Data Fig. 4d). The results showed no significant difference between these groups, suggesting that capillary EC transcytosis is unchanged in our *pdgfrb* mutants.

It might be helpful to include some orienting labels and/or additional descriptions in the figure legends to help readers who are not used to looking at zebrafish brain vessels have an easier time figuring out what they are looking at and where it is in the brain.

We thank the reviewer for this suggestion and agree that adding further information in the figure legends and illustrations about orientation would make it easier for readers. In addition to the information provided in the figure legends in the submitted version, we have added an illustration, more labels on the revised figures, extended the descriptions in figure legends, main text and methods.

We have added a schematic depicting the tracer leakage assay workflow, orientation of live imaging and analysed region of interest (Extended Data Fig. 1a–b).

All figure legends have been updated with the anatomical position and microscopy view.

Additional labels on figures have been added to understand the referenced vessel names (new data in Fig. 3c and Extended Data Fig. 4a–b′).

**Recommendations for the authors:**

**Reviewer #1 (Recommendations for the authors):**
The study uses the intensity of tracer signals within the vessels to analyze BBB permeability, potentially underestimating leakage severity. The dye intensity is measured 2 hours after injection, however, other studies have already observed leakage after 30 Minutes, by imaging directly in the brain parenchyma. The overall intensity should also decrease through leakage from the other vessels of the body, e.g. in the trunk and tail. Probably the loss of intra-vascular dye intensity from leakage in barrier-free vessels is already so high (after 2 hours) that the smaller amount of leakage across the BBB cannot be observed.

We thank the reviewer for this comment and suggestion. We agree that small sized tracers leak from vasculature, particularly through fenestrated vessels in the trunk and tail. We have based our timing on previous studies and our own experience. In zebrafish, the study by O’Brown et al 2019 also used 2 hpi[5] for detection of leakage in *mfsd2aa* mutants, which also has been proposed to regulate BBB integrity by controlling EC transcytosis. Therefore, we believe that performing experiments at 2 hpi is appropriate to investigate roles of pericytes in BBB integrity. Our data would suggest that this timing works.

In response to this and other comments, we performed further experiments and analyses to test leakage of tracers testing molecular weights ranging from 1 to 2000 kDa individually. We showed that these tracers can reliably be detected in brain parenchyma and vasculature when imaged at 2 hpi. In another study, we showed that medium size tracers such as 40 kDa Dextran can be reliably detected in the vasculature in similar timepoints[10]. Considering we have performed experiments using 10 and 70 kDa tracers do detect parenchymal tracer accumulation and tracer still within the vessels, we believe this timepoint is appropriate for assessing BBB integrity in zebrafish.

In addition to these experiments, see our tracer leakage experiments in 1-month-old animals, at 0.5 and 6 hpi to test leakage pattern described above (Fig. 5 and Extended Data Fig. 6).

Therefore, the authors will need to validate their method of choice, showing an impairment of the BBB, caused by other agents (known to affect the BBB), and at 48hpf, when the BBB is not tightened yet. One example for BBB impairment can be found in O'Brown et al (2019), eLife 8e47326. doi: 10.7554/eLife.47326

We thank the reviewer for this suggestion. As shown by O’Brown et al 2019, we have performed experiments at 3 dpf when BBB integrity is not mature and at 7 dpf when BBB is functional[5], testing both 10 and 70 kDa tracers. We detected significantly higher tracer accumulation at 3 dpf, showing our new additional method (see below) can detect tracer leakage in the brain (new data in Fig. 2e–f).

Ideally, the authors would also supplement the method with additional approaches in the younger developmental stages to validate their findings.The validation of the method and the findings is particularly important for the claims of lack of BBB impairment in the absence of mural cells, as this is a "negative" finding.

In response to this and comments from other reviewers, we performed additional tracer leakage experiments (new data in Fig. 2a–d) where we imaged 10 and 70 kDa tracers with a vascular reporter (*Tg*(*kdrl:EGFP*)*^s843^* or *Tg*(*kdrl:Hsa.HRAS-mCherry*)*^s916^*) and used this reporter for normalisation. Both this approach as well as the experiments provided in the first submission (updated as Extended Data Fig. 3a–d) showed that *pdgfrb* mutants at 7 and 14 dpf have indistinguishable BBB integrity compared to siblings. See also Author response image 1 that further addresses this.

I also strongly suggest to rephrase and downtown the claim that vascular mural cells do not control the blood-brain barrier in developing zebrafish.As a negative finding cannot be proven completely and lots of the previously shown effects on murine BBB impairment are rather weak (when caused by single agents such as Claudin5 deficiency or Sphingosine-phosphate receptor1 knockout), it might be important to only claim that in zebrafish no strong impairment (as observed in the mural cell-deficient mouse) could be observed. Or rephrase it to "no impairment as severe as/comparable to ... could be observed" and then provide an impairment control for the developmental stages.

We thank the reviewer for this comment and agree that negative findings are very challenging to prove. However, we find no evidence of leakage of the BBB in animals lacking mural cells at 7 and 14 dpf and believe that our data is robust on this point. As such, we believe we show that a vertebrate with a largely conserved EC BBB, can have intact barrier function in the absence of mural cells.

We have as suggested revised our claims throughout the manuscript to provide more further nuanced discussion of this, but we do not want to water down our claims too much as we believe they are important. We hope that the reviewer will appreciate our carefully worded and expanded discussion section.

Additional items of interest to the readers and therefore suggestions to improve the manuscript could be(1) To include more molecular analysis: while the study identifies caveolae induction and basement membrane thickening as potential contributors to focal leakage, the exact molecular mechanisms linking mural cell loss to these structural changes are not deeply investigated.(2) Also, the study primarily associates BBB disruption in the adult with aneurysms. Therefore other subtle or diffuse changes to BBB permeability that might occur even without overt vascular lesions are potentially underrepresented.However, following up experimentally on these might exceed the scope of the manuscript.

We thank the reviewer for these suggestions and agree with both points. However, as stated by the reviewer, these experiments are beyond the scope of the manuscript and represent future directions for our lab and others.

**Reviewer #2 (Recommendations for the authors):**
(1) Mouse genes should be written as follows: Pdgfb, Pdgfrb and be in italics. See line line 70: it should be written "Pdgfb and Pdgfrb (italics)" and not "PdgfB and Pdgfrβ".

We have updated the text according to the reviewer’s suggestion.

(2) Please state the age of the fish analyzed in Figure 1f and 1g.

We have moved this data to Extended Fig. 3a–d (previously Fig. 1f-g) and have placed age information on the images and in the figure legends.

(3) Is the reduced vascular complexity in pdgfb mutant due to reduced angiogenesis or due to excessive pruning?

This is a good question, and we do not know at this stage. We have unpublished data that suggest pericytes secrete angiogenic growth factors, but this question warrants a thorough investigation that we believe is beyond the scope of this current study.

(4) Please check that the figure legends state the correct number of fish analysed. For example, Figure 1 d, e N=8 but there seem to be 9 data points per group - 14dpf.

We apologise for this mistake and thank the reviewer for raising this. We have updated the graphs and figure legends accordingly.

(5) Please indicate in the figures the genotypes (wt, het) of a sibling presented alongside a pdgfb mutant.

Wild-type and heterozygous mutants are commonly used together in zebrafish research as a collective control group termed siblings. Since we didn’t see any difference between wild-type and *pdgfrbuq30bh/-* groups in any experiments, we reported these groups together. This is now stated in the supplementary materials.

One exception to this was examination of the growth and survival rates where we show the genotypes separately (new data in Extended Data Fig. 1b-f).

(6) Please explain clearly what region is shown in Figure 2B. I do not understand the explanation "approximate location of dotted line". Is the image in the panel "a" top view of a brain?

We have moved this data to Fig. 3a′ (previously Fig. 2b) and replaced the dotted line in Figure 3a (previously Fig. 2a) with a white box indicating the location of the restricted region in the whole brain image.

We have revised the text as below:

“Subset of z-slices from the whole brain imaging in (a) and (b) (white boxes) indicating mural cell loss and abnormal capillary network patterning. 100-μm-thick maximum intensity projections (MIP) were generated using the continuation of the left middle mesencephalic central artery (MMCtA, arrow) as an anatomical landmark.”

In addition, we have updated all our figure legends clearly stating the view and anatomical position of the imaged sample.

(7) Figure 2e: Note that- the dotted areas do not correspond to the areas magnified. Please adjust.

We have moved this data to Extended Data Fig. 5a (previously Fig. 2e–e′) and updated the location of the white box in 5a shown in enlarged view in 5a′.

(8) Lines 112 and 114 - Should the indicated figure be Figure 2b-d and Figure 2c-d, respectively, and not Figure 1?

We thank the reviewer for pointing out this mistake. All the figure legends are now referred to appropriately in the revised manuscript.

(9) Data presented in Figure 2 and Figure 3 can be consolidated and presented as one Figure.

We thank the reviewer for this suggestion. After addition of new data and revising the manuscript we have decided to keep these data presented separately.

(10) Note that Figure 2a,b shows 5-month-old fish, not 2-month-old fish. Additionally, Extended Data Figure 3 shows 5-month-old fish, not 3-month-old fish.

The stages noted by the reviewer were correctly indicated.

(11) Figure 2d: Please clarify the definition of a "large vessel".

We have observed normal morphology in capillaries and noted aneurysms and hotspots in large calibre vessels such as arteries, which become more severe over time. We have revised this across the manuscript accordingly.

(12) Figure 4a, b: Please explain how the hotspots of leakage were defined based on the extravasated tracer.

Hotspots of leakage are scored when fluorescent tracer aggregates are clearly observed outside the vessels. Vessel borders were defined using the transgenic lines (*Tg*(*kdrl:EGFP*)*^s843^* or *Tg*(*kdrl:Hsa.HRAS-mCherry*)*^s916^*). We have added a clear description in the methods section (lines 473–475).

Figure 4c: Why were Pdgfrb crispants used and not the mutant line?

They were used as *pdgfrb* crispants phenocopy the lack of brain mural cells (Extended Data Fig. 5e, previously Extended Data Fig. 4b) and mutant phenotype reliably and for practical reasons, because they allow faster experiments and reduce fish usage.

Figure 4e: The magnification of the electron microscopy images does not make it possible to clearly identify caveolae. What was the magnification of the collected images for caveolae analysis? How did the authors ensure that they quantified only caveolae and not other types of vesicles?

Respectfully, we disagree that the magnification is insufficient as our images were captured and analysed consistent with previous ultrastructural descriptions[11,12]. We based our quantification of caveolae on the size of vesicles observed and define them as circular profiles of less than 100 nm in diameter and were scored as luminal or abluminal based on proximity to each surface membrane (within 500 nm of each surface or in a thin-walled vessel the caveolae closest to each surface) (lines 398–409). Importantly, comparable analyses at similar magnifications have been independently validated in multiple caveola-deficient zebrafish genetic models[4,13]. Interestingly given the reviewers comments above, we do see increased vesicular structures that are larger than caveolae, but we only provide quantification of the caveolae here.

**Reviewer #3 (Recommendations for the authors):**
Congratulations to the authors on their really beautiful imaging and rigorous quantitative documentation of phenotypes - this is a really nicely done study, and could be very important to the field with just a few additional experiments to buttress the key conclusions.

We thank the reviewer for their kind comments.

In addition to the comments noted in the public review, I would only point out that there are two mislabeled call-outs in the text (Lines 112 and 114; says Figure 1, should say Figure 2).

We thank the reviewer for this point and have now revised the text accordingly.

(1) Ando, K., Ishii, T. & Fukuhara, S. Zebrafish Vascular Mural Cell Biology: Recent Advances, Development, and Functions. Life (Basel) 11 (2021). https://doi.org/10.3390/life11101041

(2) Ando, K. et al. Clarification of mural cell coverage of vascular endothelial cells by live imaging of zebrafish. Development 143, 1328-1339 (2016). https://doi.org/10.1242/dev.132654

(3) Ando, K. et al. Conserved and context-dependent roles for pdgfrb signaling during zebrafish vascular mural cell development. Dev Biol 479, 11-22 (2021). https://doi.org/10.1016/j.ydbio.2021.06.010

(4) Lim, Y. W. et al. Trans-Endothelial Trafficking in Zebrafish: Nanobio Interactions of Polyethylene Glycol-Based Nanoparticles in Live Vasculature. ACS Nano (2026). https://doi.org/10.1021/acsnano.5c21042

(5) O'Brown, N. M., Megason, S. G. & Gu, C. Suppression of transcytosis regulates zebrafish blood-brain barrier function. Elife 8 (2019). https://doi.org/10.7554/eLife.47326

(6) O'Brown, N. M. et al. The secreted neuronal signal Spock1 promotes blood-brain barrier development. Dev Cell 58, 1534-1547 e1536 (2023). https://doi.org/10.1016/j.devcel.2023.06.005

(7) Armulik, A. et al. Pericytes regulate the blood-brain barrier. Nature 468, 557-561 (2010). https://doi.org/10.1038/nature09522

(8) Daneman, R., Zhou, L., Kebede, A. A. & Barres, B. A. Pericytes are required for blood-brain barrier integrity during embryogenesis. Nature 468, 562-566 (2010). https://doi.org/10.1038/nature09513

(9) Mae, M. A. et al. Single-Cell Analysis of Blood-Brain Barrier Response to Pericyte Loss. Circ Res 128, e46-e62 (2021). https://doi.org/10.1161/CIRCRESAHA.120.317473

(10) Lim, Y.-W. et al. A Standardized Protocol to Investigate Trans- Endothelial Trafficking in Zebrafish: Nano-bio Interactions of PEG-based Nanoparticles in Live Vasculature. bioRxiv, 2025.2007.2023.666282 (2025). https://doi.org/10.1101/2025.07.23.666282

(11) Parton, R. G. & Simons, K. The multiple faces of caveolae. Nat Rev Mol Cell Biol 8, 185-194 (2007). https://doi.org/10.1038/nrm2122

(12) Parton, R. G. & del Pozo, M. A. Caveolae as plasma membrane sensors, protectors and organizers. Nat Rev Mol Cell Biol 14, 98-112 (2013). https://doi.org/10.1038/nrm3512

(13) Lim, Y. W. et al. Caveolae Protect Notochord Cells against Catastrophic Mechanical Failure during Development. Curr Biol 27, 1968-1981 e1967 (2017). https://doi.org/10.1016/j.cub.2017.05.06